# Targeting HIV-1 conserved regions: An immunoinformatic pathway to vaccine innovation for the Asia

Akmal Zubair [1]*, Ahmed Al-Emam[2], Muhammad Ali[1], Syeda Maryam Hussain[3], Ranya Mohammed Elmagzoub[4]

1 Department of Biotechnology, Quaid-i-Azam University Islamabad, Pakistan, 2 Department of Pathology, College of Medicine, King Khalid University, Asir , Saudi Arabia, 3 Department of Livestock Production and Management, Faculty of Veterinary and Animal Sciences PIR Mehr Ali Shah-Arid Agriculture University Rawalpindi, Shamsabad, Murree Road, Pakistan, 4 Department of Medical Laboratory Technology, Faculty of Applied Medical Sciences, Northern Border University, Arar, Saudi Arabia

* akmalkhattak1994@gmail.com

## Abstract

A combination of humoral and cell-mediated immune system stimulation is essential for developing an effective HIV vaccine. Traditional treatment options and the challenges posed by drug resistance necessitate the discovery of a viable vaccine candidate capable of eliciting a robust immunological response. This research aims to develop an HIV vaccine with a multi-epitope component using a unique immunoinformatics approach. A subunit vaccine comprising B-cell, helper T-cell, and cytotoxic T-cell epitopes, along with appropriate adjuvants and linkers, was employed to identify conserved regions in the Pol, Vpr, Gag, Tat, Env, Nef, and Vif proteins. The HIV subunit vaccine demonstrated the potential to activate both cell-mediated and humoral immune responses, indicating its immunogenicity. The application of homology modeling and refinement further enhanced the model's accuracy. Subsequently, the molecular docking procedure utilized the refined model structure to bind to the immunological receptor TLR-3 in lymphocyte cells. Following this, the potential interactions of the subunit vaccine with TLR-3 were investigated using molecular dynamics modeling. The vaccine's stability was improved through a meticulous disulfide engineering technique that involved inserting cysteine residues into highly flexible regions. Finally, in silico cloning was employed to validate the efficacy of translating and producing the vaccine in a microbiological setting. The vaccine shows promising results in terms of population coverage, reaching 82% of the global population, with extraordinary efficacy in Asia, covering up to 95% of the population. Our HIV vaccine candidate is highly stable and elicits a robust immune response against HIV-1.

## Introduction

There were around 690,000 fatalities that occurred as a direct consequence of HIV infections in 2019, with 1.7 million individuals infected [1]. The persistence of this condition is notable, despite the availability of drugs. It is widely believed that an effective vaccine is essential to eradicate HIV infection and its associated disease, acquired immunodeficiency syndrome

**Data availability statement:** All relevant data are available within the paper and its Supporting Information files.

**Funding:** The author(s) received no specific funding for this work.

**Competing interests:** The authors have declared that no competing interests exist.

(AIDS) [2]. This scenario has occurred despite the fact that these data were 23% lower than those recorded in 2010. Additionally, various effective treatments are available, including dolutegravir, efavirenz, and ritonavir [3]. Because the COVID-19 pandemic is still ongoing, it has become evident that the only vaccine capable of significantly reducing the infection rate, as well as the morbidity and mortality associated with COVID-19, is both effective and widely distributed. Therefore, this vaccine must be manufactured [4,5]. A significant number of individuals believe that the primary reasons for the inadequate regulation of HIV medications are their high cost, limited availability, and perceived ineffectiveness. This perspective is shared by many. Patients who are actively engaged in the process of receiving antiretroviral therapy [6]. Individuals are developing increasing resistance to the medications they are currently using, which is one of the significant factors contributing to the rising number of HIV infections worldwide. Regarding the acceleration of AIDS transmission on a global scale, it is widely accepted that HIV-1, the more prevalent and lethal of the two strains of HIV, is the primary agent responsible for the rapid spread of the disease [7,8]. Controllers are a specific subgroup of HIV-positive individuals who can remain asymptomatic for years while maintaining a high CD4 + cell count without the use of antiretroviral medication, which is typically required for most. These individuals experience these benefits without the need for such treatment. Among the population of HIV-positive individuals, controllers represent the highest echelon of this group [9–11]. However, the elicitation of broadly neutralizing antibodies is the most crucial component in defending against AIDS. This remains true despite the absence of immunological correlations regarding the efficacy of non-neutralizing antibodies against HIV-1 infection. Furthermore, it has been demonstrated that the activation of CD4 + and CD8 + T cells is advantageous in the fight against HIV infection, a fact that has been recognized for quite some time [6,12].

An HIV-1 therapeutic vaccine is being developed to stimulate a more specific and robust immune response that targets conserved viral epitopes. This approach aims to enhance the immunological responses that the body generates in reaction to a natural infection [13]. This will enhance the immune responses that the body generates as a result of inflammation. Consequently, the pursuit of viable candidates for therapeutic vaccines against HIV-1 has been hindered by challenges related to delivery systems and immunogen design. As a result, the search for these vaccines has faced significant obstacles [14]. The development of therapeutic vaccines for HIV-1 is being advanced by the production and assembly of synthetic multiepitope immunogens, which represent a promising option [14,15]. Common viral antigens served as the foundation for the development of these immunogens, which were designed based on a diverse array of immunostimulatory, defensive, and T-cell epitopes. The primary objective is to rejuvenate the immune system, enhancing its ability to combat the effects of HIV-1 infection more effectively [16]. Using this strategy, the objective is to enhance therapeutic vaccines against HIV-1 by overcoming previously established limitations [17,18]. In addition, in silico methods are employed to facilitate the identification of epitopes—peptides that have the potential to be immunogenic—within the linear protein sequence. Docking and molecular dynamics are two strategies that can be applied in relation to the major histocompatibility complex (MHC), which comprises MHC-I and MHC-II [19–21]. An investigation into the varying degrees of affinity and stability exhibited by the peptide-protein complex is the method through which this objective may be achieved. Rodriguez-Fonseca and his colleagues conducted research on the dendrimer-G4-PAMAM-peptide complexes. This study was facilitated by three-dimensional models of HIV-1 gp120 [22–26]. Because of its significant role in both the fusion of the virus to the host and the potentially infectious nature of HIV-1, the development of an HIV-1 vaccine relies heavily on gp120 as its principal antigen. Our research primarily focused on creating effective strategies to elicit cytotoxic T lymphocyte

(CTL) and helper T lymphocyte (HTL) specific T-cell responses. This was the central aim of our investigation. The in-silico design process we developed is an efficient technology that could potentially be utilized in the production of a recombinant vaccine [27]. By researching the conserved regions and selecting the optimal CTL and HTL epitopes that can activate B cell responses, we successfully achieved our goal. This was followed by the construction of the vaccine design, which incorporated the Gb-1 adjuvant and appropriate linkers to enhance the induction of cellular immunity There are several Insilco vaccines designed for HIV. These vaccines all utilize a single protein-based approach for vaccine development. Due to the high rate of mutations in the HIV genome, these vaccines may be compromised upon the mutation of that region, but in our vaccine structure, we used the conserved region in the HIV genome, which is less prone to mutation and possibly sustained for a longer period

The aim of this study is to design a novel stable in-silico vaccine for HIV that can the majority region of Asia. There are several in-silico vaccines developed for HIV, but all these vaccines use a single protein or partial protein as a vaccine candidate. We came up with the novel idea that weather-conserved regions in HIV genomics can act as stable vaccine candidates. For this purpose, the conserved region in the HIV genome was identified and used as an HIV vaccine candidate.

## 2. Material and methodology

### 2.1 Obtaining genomic data

Genome and proteome data are now easily accessible due to the abundance of databases. Researchers can gather crucial information about microorganisms from various species, including humans, through several proteome and genome libraries. Notable websites facilitate data retrieval by providing complete sequences and detailed information. In this context, the National Center for Biotechnology Information (NCBI) stands out among other such databases [28]. 61 whole genome data was retrieved from NCBI (https://www.ncbi.nlm.nih.gov/). Using BioEdit (BioEdit 7.2) and Clustal Omega (https://www.ebi.ac.uk/jdispatcher/msa/clustalo) for multiple sequence alignment and identified the conserved sequences. These conserved regions were further used for vaccine development.

### 2.2 B cell epitope predictions

The ABCpred server was utilized to gather predictions regarding specific epitopes associated with the humoral immune response for the provided HIV proteins, primarily focusing on linear B-cell epitopes [29]. To predict B-cell epitopes, this service employs an artificial neural network. The server utilizes a recurrent neural network that analyzes a dataset consisting of 700 B-cell epitopes and 700 non-B-cell epitopes. It is important to note that the epitopes in question have a maximum length of twenty residues and are used for both preparation and evaluation purposes. The recurrent neural network employed by this server has demonstrated an accuracy rate of 65.93%. Based on this, the FASTA sequences of entire HIV proteins were used to assess linear B-cell epitopes, applying a threshold of 0.51.

### 2.3 CTL and HTL epitopes predication

The NetCTL-1.2 server (https://services.healthtech.dtu.dk/services/) was used in order to make predictions about the CTL epitopes that were present in the viral proteins [30,31]. It was determined that the TAP efficiency parameter would be set at 0.05, the C-terminus cleavage parameter at 0.15, and the threshold at 0.75. The epitope prediction process also involved the preselection of additional parameters. Selecting these parameters was essential to ensure accuracy in the analysis. To gather scores for peptide attachment and proteasomal C-terminal

cleavage with MHC-I, an artificial neural network was employed. This facilitated the collection of relevant ratings for the study. Throughout this process, the TAP efficiency score was calculated using a weight matrix method. The IEDB online service provides various features, including the ability to predict epitopes for helper T lymphocytes (HTL). To evaluate the methodology, a reference set consisting of seven human leukocyte antigens (HLAs) was utilized. A comprehensive analysis demonstrated that the server's blind epitope predictions were accurate for each of the seven reference HLAs. Utilizing data from the Information Exchange Database (IEDB), an inverse correlation was identified between the IC50 values of the predicted epitopes and the binding affinity of MHC-II. For MHC-II, a high binding affinity is observed for IC50 values less than 50 nM, a medium affinity for values less than 500 nM, and a low affinity for values exceeding 500 nM. Specifically, if the IC50 value exceeds 500 nM, the binding affinity is considered low.

### 2.4 Protein antigenicity allergenicity and toxicity

The antigenicity of a protein or peptide sequence can be assessed using Vaxijen, a web-based application that simplifies the process. VaxiJen may be found at https://www.ddg pharmfac. net/vaxijen/VaxiJen/VaxiJen.html [32]. Several factors are considered by the AllergenFP server to estimate allergenic peptides. In addition to hydrophobicity, peptide size, the formation of helical structures and beta-strands, and the frequency of each amino acid residue, these characteristics play a crucial role. AllergenFP can be examined at https://ddg-pharm-fac.net/index.html. Leading the list is the ToxinPred server, which identifies potentially hazardous peptides. This tool employs a machine learning approach known as Support Vector Machines (SVM) to predict substances that may be harmful. It also takes into account a quantitative matrix. The ToxinPred server can be found at https://webs.iiitd.edu.in/raghava/toxinpred/1.

### 3.5 Population coverage

The IEDB servers were used for population coverage. The MHC-1 and MHC-II coverage was identified separately and combined.

### 2.6 multi-epitope vaccine construction

A variety of distinct linkers were employed to establish connections between the selected epitopes. Specifically, AAY linkers were used to join cytotoxic T lymphocyte (CTL) epitopes, while GPGPG linkers were utilized to connect helper T lymphocyte (HTL) epitopes. Additionally, to enhance the immunogenicity of the vaccine, an EAAAK linker was incorporated to connect the vaccine sequence to the N-terminal. This approach was implemented to significantly boost the vaccine's immunogenicity. The design process of the HIV vaccine specifically utilized these linkers due to their ability to improve epitope presentation, thereby enhancing protection against HIV. The underlying rationale for this strategy was to facilitate a more effective immune response [33]. Furthermore, the linkers effectively prevent any folding or aggregation by maintaining a clear separation between the epitopes.

### 2.7 Physiochemical properties of HIV vaccine

The characteristics of the HIV vaccine candidate were assessed utilizing the ProtParam web server, accessible at https://www.expasy.org/protparam/ [34]. The aliphatic index, molecular weight, and isoelectric point (pI) were thoroughly investigated and assessed throughout the course of the experiment. Furthermore, it is imperative to consider the half-lives of chemicals both within living organisms and in controlled laboratory environments. Additionally, it is crucial to account for the sensitivity index and the amino acid composition.

## 2.8 Modelling of secondary structure

The PSIPRED v3.3 analytical workbench was used to determine the secondary structure of the vaccine anticipated for market release. Subsequent calculations were conducted to quantify the relative proportions of helices, strands, and coils [29].

## 2.9 Tertiary structure modeling

The construction of the three-dimensional (3D) tertiary model of the multi-epitope vaccine was performed using I-TASSER, which stands for Iterative Threading ASSEmbly Refinement. You can access this server by visiting the website https://zhanglab.ccmb.med.Umich.Edu/I-TASSER/ which is located online. To produce accurate predictions regarding the function and structure of proteins, the I-TASSER server serves as a comprehensive platform that utilizes computational methods. This research employed a methodology based on the sequence-to-structure-to-function framework, utilizing data from the Protein Data Bank (PDB). The aim of this study was to identify structural patterns that are comparable to those found in other protein structures, as well as to uncover patterns relevant to various protein configurations. I-TASSER employs multiple threading alignments and iterative structure-building techniques to generate three-dimensional atomic models. The process begins with an amino acid sequence as the initial input. A template modeling (TM) value exceeding 0.5 indicates that the model has a correct topology, while a TM value below 0.17 suggests that any observed similarity is coincidental rather than meaningful. The length of the protein does not affect the selected cutoff value. I-TASSER has been recognized as the most efficient server for analyzing the three-dimensional structure of proteins, as demonstrated by its performance in the five CASP (Critical Assessment of Methods of Protein Structure Prediction) evaluations conducted within the scientific community over the past five years [35].

## 2.10 Refinement of tertiary structure and validation

The process under discussion involves the validation and verification of the three-dimensional structure of atoms within a molecule. In the context of generating the three-dimensional configuration of the final vaccine structure, accomplished using the RaptorX server, the degree of similarity between various alternative template structures and the target is a critical consideration. Consequently, the accuracy of the model structure was improved beyond its initial level through the application of the GalaxyRefine server [36]. Beginning with the reconstruction of the side chain, GalaxyRefine utilizes molecular dynamics modeling to perform side chain packing and overall relaxation. Highly developed methods for predicting the tertiary structure of proteins have the potential to enhance the accuracy of structural model predictions both locally and globally. Additionally, the Ramachandran plot was analyzed using the PDBsum server, an essential tool for assessing the reliability of protein structures [37]

## 2.11 Disulphide bond identification

This study was conducted with the primary objective of enhancing the stability of a model vaccine. Utilizing a systematic approach, the goal was successfully achieved by increasing the production of disulfide bonds. Distinct disulfide bonds were created within the vaccine's structure. The ability of disulfide bonds to attain optimal geometric conformation and provide significant stability to protein structures is a well-documented characteristic of molecules containing these bonds. Disulphide by Design-2 (DbD2) is an essential online tool for constructing the framework of the model vaccine. The DbD2 approach is based on identifying residue pairings with a high propensity to form disulfide bonds when individual amino acids are replaced with cysteine. The outcome of this process is a collection of residue pairs that exhibit favorable geometry and the potential to form disulfide bonds [38].

## 2.12 Molecular docking between HIV vaccine construct and TLR3 and TLR5

To perform docking between the proposed tertiary enhanced HIV vaccine construct, which includes Gag, Pol, Vif, Vpr, Tat, and Nef, and TLR3 (PDB ID: 1ziw), we utilized the HDOCK server. This approach was employed to determine the effectiveness of the vaccination. Selecting the TLR3 receptor was a logical choice due to its well-established role in the immune system, which was a key factor in our decision-making process [39].The system has the ability to recognize a wide range of infectious agents and to activate robust immune responses. Understanding the stimulation of signaling pathways involved in the production of cytokines and interferons is crucial. The complex mechanisms employed by proteins, along with the energy associated with the binding process, inspired the development of this docking server. Within a timeframe of less than two hours, the Hdock server successfully generated ten docking structures, with each cluster optimally organized based on the parameter exhibiting the lowest energy. This methodology offers various possibilities, including the selection of multiple energy factors, the creation of additional restraint files, and the execution of a comprehensive analysis of the results.

## 2.13 Molecular docking between T-cell epitopes and HLAs

To perform molecular docking between HLA alleles and T-cell epitopes, the https://data-science.unm.edu/tomcat/biocomp/convert tool was used to convert the T-cell epitopes into pdb format. The respective HLA allele was retrieved from the protein data bank. Discovery Studio performed a refinement step before docking. The HDOCK server was used for molecular docking between T-cell epitopes and HLA alleles.

## 2.14 Simulation-based computational approach

The web server known as CImmSim utilizes an agent-based modeling approach as its foundation. You can access the website at the following address: https://kraken.iac.rm.cnr.it/C-IMMSIM/. To evaluate the experiment's results, the effects of a foreign antigen particle on the immune system were examined through simulations. The PSSM approach is utilized to illustrate how an antigen triggers an immune response. Following the administration of the vaccine, the server conducts an analysis of the production of interferon, cytokines, and antibodies. Typically, there is a four-week interval between the routine doses of vaccines and the subsequent doses. During the time intervals ranging from one to eighty-four hours, measured in eight-hour increments, each parameter remained consistent with its initial state. A four-week gap was maintained between each of the two shots administered in this specific series of events. The web server was employed to predict the responses of Th1 and Th2 cells. Default settings were used to calculate the Simpson Index, commonly referred to as D.

## 2.15 Optimization of sequences and virtual cloning

Codon adaptation is a technique that modifies codon usage to align with the preferences of prokaryotic bacteria commonly encountered during sequencing. This approach can lead to an increased rate of gene expression (https://en.vectorbuilder.com/tool/codon-optimization.html). The primary objective of this method is to enhance the production rate of the vaccine protein in the E. coli K12 host by leveraging the differences in codon usage between the selected host organism and humans. Among the available options, we selected three key factors: the ribosomal binding site in prokaryotes, the cleavage sites for restriction enzymes, and the suppression of rho-independent transcription termination. To assess whether the optimized codon sequence contained any commercially viable restriction enzyme recognition

sites, specifically BstZ17I and Eco53KI, an analysis was conducted. According to the findings, there was no evidence of either of these locations. As a result, the vaccine protein's optimal sequence was reversed and altered to add BstZ17I and Eco53KI restriction sites at the N- and C-terminal positions, respectively, to ensure. The modified sequence was then integrated into the pET28a(+) vector using the SnapGene restriction cloning module, with the insertion occurring at a precise location between the Eco53KI (188) and BstZ17I (2978) sites [40].

### 2.16 IMODS MD simulation

The vaccine design that showed the most promising results in the molecular docking study involving the HIV vaccine structure and the TLR3 receptor underwent a molecular dynamics simulation analysis. This analysis was conducted using the iMODS web server, which can be accessed at http://imods.chaconlab.org/. This server is well-known for its rapid performance, continuous availability, and strong capabilities in computing and evaluating protein flexibility [41].

### 2.17 MD simulation of Vaccine construct and HIV-TLR3 complex

To conduct molecular dynamics simulations on the vaccine construct, GROMACS version 4.6.5 was utilized. Initially, the vaccine design was neutralized and then fully hydrated with water molecules, resulting in a reduction in energy consumption. During the NVT and NPT optimizations, the temperature was maintained at 300 K, the pressure at 1 bar, and the constraint forces were set to 1000 kJ/mol. Subsequently, the duration of the molecular dynamics simulation was one hundred nanoseconds.

## 3. Results

### 3.1 Conserved region identification

The conserved sequences were identified in the Pol, Vpr, Vif, Env, Gag, Tat, and Nef regions. A total of 616 conserved base pairs were found across the Vif, Pol, Vpr, Tat, Gag, Nef, and Env regions. Table 1 illustrates the conserved regions in the various genes at different positions. The ExPASy Translate tool was utilized to determine the protein's primary structure (https://web.expasy.org/translate/).

### 3.2 B cell epitopes prediction

The IEDB server identified six B cell epitopes. The analysis of antigenicity, toxicity, and allergenicity indicates that each epitope is a promising antigen, with no associated toxicity or allergenicity, as illustrated in Figs 1A and 1B

We have also identified the discontinuous epitopes of B cells. The discontinuous B cell epitopes are represented in Fig 2.

### 3.3 CTL and HCL epitopes prediction

To develop the vaccine, the researchers sought epitopes with a high affinity for cytotoxic T lymphocyte (CTL) receptors and a low percentile rating for helper T lymphocyte (HTL) receptors. Before finalizing the selection of HTL epitopes, several additional factors were considered. These included the ability to induce interferon (IFN), the absence of allergenicity (with a score below -0.4), and the lack of overlapping sequences. The capacity of the HTL epitopes to induce IFN and their non-allergenic properties were predicted using the Algpred and IFNepitope servers. Seventeen MHC-I epitopes were selected. Similarly, 17 MHC-II epitopes were chosen based on percentile score, toxicity, allergenicity, antigenicity, and solubility.

**Table 1. Represents different conserved in various genes of HIV along with other positions in the genome.**

| Gene Name | Conserved regions and sequences | Position in | Size |
|---|---|---|---|
| Gag region | GGAGCCACCCCACAAGATTTAAA | 1568 to 1590 | 22 bp |
| | ATAGCAGGAACTACTAG | 1742 to 1758 | 16 bp |
| | AAAATAGTAAGAATGTATAGCCCT | 1850 to 1873 | 23 bp |
| | AGACAGGCTAATTTTTTAGG | 2324 to 2343 | 19 bp |
| | TCCCTCAAATCACTCTTTGGCA | 2528 to 2549 | 21 bp |
| Pol | ACAGGAGCAGATGATACAGT | 2605 to 2624 | 19 bp |
| | CCAATTAGTCCTATTGA | 2827 to 2843 | 16 bp |
| | CCAGTAAAATTAAAGCCAGG | 2851 to 2870 | 19 bp |
| | TTAAACAATGGCCATTGACAGAAGA | 2888 to 2912 | 24 bp |
| | AAATCAGTAACAGTACT | 3136 to 3152 | 16 bp |
| | CCACAGGGATGGAAAGG | 3274 to 3290 | 16 bp |
| | AGCATGACAAAAATCTT | 3313 to 3329 | 16 bp |
| | AGCTGGACTGTCAATGA | 3577 to 3593 | 16 bp |
| | TGGACATATCAAATTTATCA | 3835 to 3854 | 19 bp |
| | GTCAATACCCCTCCT | 4075 to 4089 | 14 bp |
| | TATGCATTAGGAATCATTCA | 4329 to 4348 | 19 bp |
| | TCATGGGTACCAGCACA | 4428 to 4444 | 16 bp |
| | TGTGATAAATGTCAG | 4626 to 4640 | 14 bp |
| | | Genome | |
| Vif | ATCCCAGCAGAAACAGG | 4773 to 4789 | 16 bp |
| | GGAATTCCCTACAATC | 4926 to 4942 | 16 bp |
| | ATGGCAGTATTCATTCACAATTTTAA AAGAAAAGGGGGGATTGGGGGGTACAGTGCAGG | 5040 to 5098 | 58 bp |
| | GAAAGAATAATAGACAT | 5100 to 5116 | 16 bp |
| VPR | CAGGGACAGCAGAGA | 5189 to 5203 | 14 bp |
| | ATTTGGAAAGGACCAGC | 5208 to 5224 | 16 bp |
| | TGGAAAGGTGAAGGGGCAGT | 5235 to 5254 | 19 bp |
| Tat | AGCAGAATAGGCATT | 6072 to 6086 | 14 bp |
| | TCCTATGGCAGGAAGAAGCGGA | 6253 to 6274 | 21 bp |
| Env | AATTGGAGAAGTGAA | 8115 to 8129 | 14 bp |
| | GGAAGCACTATGGGCGC | 8265 to 8281 | 16 bp |
| | GTCTGGGGCATTAAACA | 8394 to 8410 | 16 bp |
| Nef | GGATCAACAGCTCCT | 8450 to 8464 | 14 bp |
| | CTCATCTGCACCACTA | 8490 to 8505 | 15 bp |
| | CTTTTTAAAAGAAAAGGGGGGACTGGA | 9591 to 9617 | 26 bp |

All selected epitopes are water-soluble, possess favorable antigenic scores, and are free from toxicity and allergenicity. The percentile score for MHC-I is 0.1, while for MHC-II, it is 1. All selected CTL and HTL epitopes are presented in Tables 2 and 3.

### 3.4 *Protein* antigenicity, toxicity, and allergenicity

The protein antigenicity score for the Pol, Vpr, Gag, Vif, Tat, Nef, and Env regions is 0.782615, indicating a strong antigenic character. The threshold value for antigenicity is 0.4. Analysis from the ToxinPred server shows that the peptides are neither as toxic as the selected peptides nor allergenic, as illustrated in Tables 2 and 3. Following the construction of the vaccine

A

**Predicted peptides:**

| No. | Start | End | Peptide | Length |
|---|---|---|---|---|
| 1 | 4 | 17 | ERQEPPHKILYTRN | 14 |
| 2 | 20 | 20 | K | 1 |
| 3 | 22 | 44 | VRMYSPRQANFLGPSNHSLATGA | 23 |
| 4 | 58 | 94 | KARLNNGHCQKKISNSTPQGWKGASQKSYLDCQSWTY | 37 |
| 5 | 110 | 116 | IMGTSTC | 7 |
| 6 | 118 | 219 | KCQIPAETGEFPTIHGSIHSQFYKKRGDWGVQCRERIIDIRDSREFGKDQLERSRGSSRIGISYGRKKRKLEKSRKHYGRVWGIKQDQQLLSSAPLLFKRKG | 102 |

B

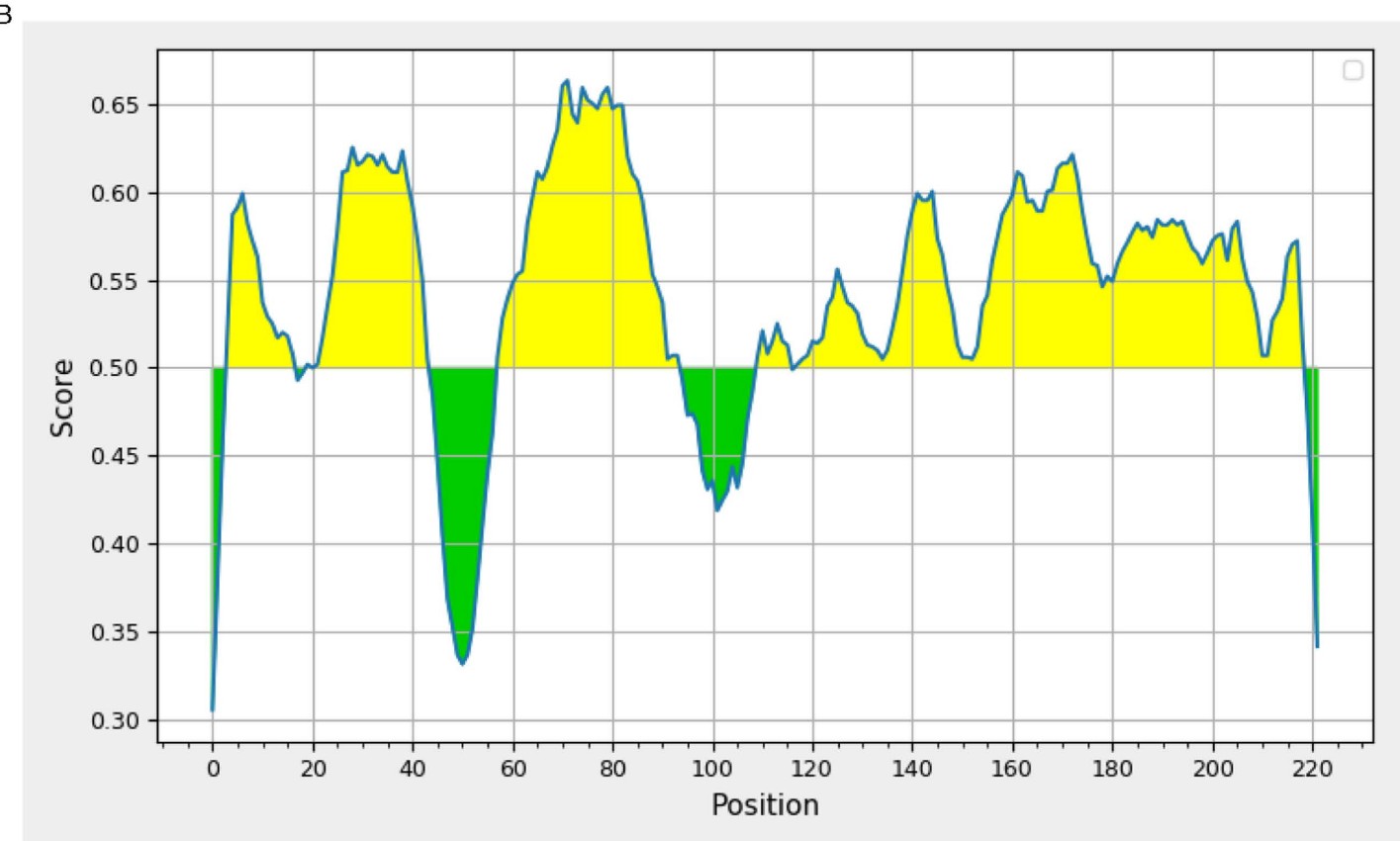

**Fig 1.** 1A Represents the selected B-cell epitopes and 1B represents the residues of yellow color which are above the threshold level were predicted as epitopes.

structure, the toxicity, allergenicity, and immunogenicity were re−evaluated. The antigenicity of our vaccine construct remains at 0.782615, while the allergenicity score is −1.290302. Additionally, the solubility results indicate that the vaccine construct is highly soluble in water, with a solubility score of 0.995973.

## 3.5 Population coverage

The IEDB population coverage server was utilized to assess the degree of population coverage for the selected epitopes. Census data indicates that approximately 80% of the global population carries MHC−I and MHC−II alleles. A population coverage scan was conducted, resulting in the acquisition of this information. With a high likelihood of effectiveness in the Asia region, the following countries exhibited the highest epitope coverage: Taiwan (95.69%), Japan

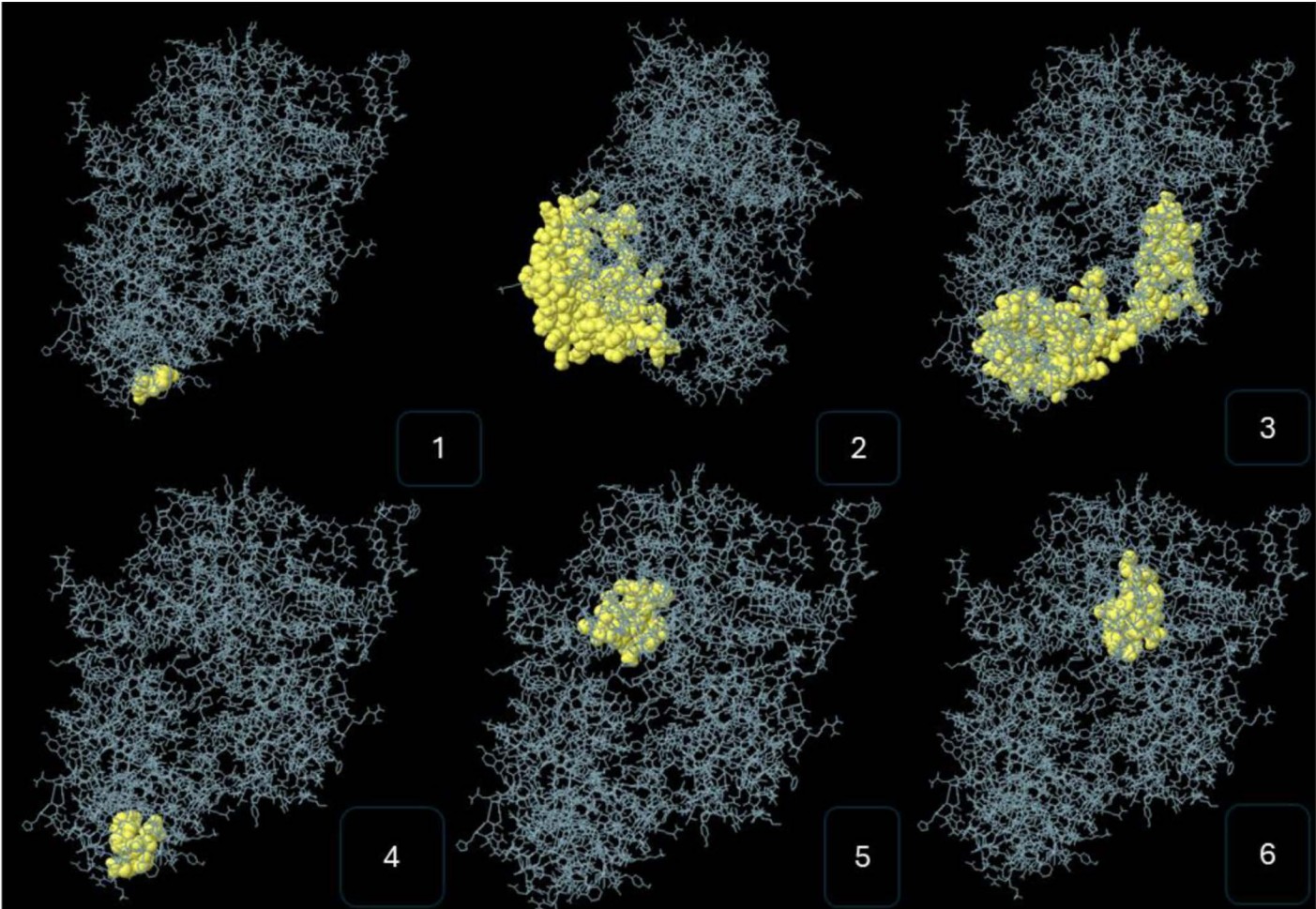

**Fig 2. Represent the discontinues epitopes of B cell on 3D structure of HIV vaccine.**

(93.49%), South Korea (90.87%), Hong Kong (90.69%), Thailand (88.7%), China (88.57%), Vietnam (87.16%), Singapore (87%), Thailand (86.77%), Pakistan (81.42%), and the Philippines (81.2%). A significant portion of the world's population has inherited the target MHC alleles, as demonstrated in the population coverage research. Consequently, we are optimistic that our vaccine may effectively curb the global spread of the virus. The population covered by HIV vaccines is detailed in Table 4.

## 3.6 Vaccine construction

The final vaccination sequence was constructed by connecting the selected HTL, CTL, and B cell epitopes using various linkers, such as AAY and GPGPG KK. An adjuvant is incorporated into the multi−epitope vaccine to elicit a more robust immune response. To improve the vaccination process, the non−toxic human beta defensin−2 (hBD−2) variant was linked to the N−terminus using an EAAAK linker. We used different linkers such as GPGPG, KK, and AAY. Linkers such as GPGPG and KK were used to enhance the immunogenicity of vaccines. The linker AAY was used to reduce steric hindrance and allow the epitopes to properly fold. They enhance receptor epitope presentation and inhibit junctional epitope synthesis by acting as connectors. In this investigation, we

**Table 2. Represents the number of selected epitopes for MHC-II based on Antigenicity, Solubility, Allergenicity, and Toxicity.**

| HTLs | MHC –II Alleles | Solubility | Allergenicity | toxicity | antigenicity score |
|---|---|---|---|---|---|
| CRERIIDIRDSREFG | HLA−DRB1*04:05 | Water soluble | −0.287 | −0.93 | 0.9237 |
| ERIIDIRDSREFGKD | HLA−DRB1*04:05 | | −0.239 | −1.22 | 0.5535 |
| GISYGRKKRKLEKSR | HLA−DRB1*13:02 | | −0.338 | −1.09 | 0.8248 |
| GSSRIGISYGRKKRK | HLA−DRB5*01:01 | | −0.290 | −0.86 | 1.4548 |
| IGISYGRKKRKLEKS | HLA−DRB1*13:02 | | −0.332 | −1.04 | 1.3324 |
| IHSQFYKKRGDWGVQ | HLA−DRB1*08:02 | | −0.125 | −1.11 | 1.3588 |
| ISYGRKKRKLEKSRK | HLA−DRB1*13:02 | | −0.279 | −1.01 | 0.8442 |
| KQDQQLLSSAPLLFK | HLA−DRB1*09:01 | | −0.224 | −0.98 | 0.4649 |
| QDQQLLSSAPLLFKR | HLA−DRB1*09:01 | | −0.227 | −0.94 | 0.5829 |
| RERIIDIRDSREFGK | HLA−DRB1*04:05 | | −0.293 | −0.87 | 0.8957 |
| RIGISYGRKKRKLEK | HLA−DRB1*13:02 | | −0.330 | −1.25 | 1.1025 |
| RIIDIRDSREFGKDQ | HLA−DRB1*12:01 | | −0.252 | −0.80 | 0.7077 |
| RNKVRMYSPRQANFL | HLA−DRB1*15:01 | | −0.270 | −1.05 | 0.4588 |
| SSRIGISYGRKKRKL | HLA−DRB5*01:01 | | −0.335 | −0.97 | 1.5385 |
| TRNKVRMYSPRQANF | HLA−DRB1*15:01 | | −0.319 | −0.93 | 0.5177 |
| VQCRERIIDIRDSRE | HLA−DRB1*04:05 | | −0.25 | −0.84 | 1.3787 |
| VWGIKQDQQLLSSAP | HLA−DRB1*03:01 | | −0.079 | −1.39 | 0.4298 |

**Table 3. Represents the number of selected epitopes for MHC−I based on Allerginicity Solubility, Antigenicity and Toxicity.**

| CTLs | HLA Alleles | water solubility | allergenicity | toxicity | antigen score |
|---|---|---|---|---|---|
| ISNSTPQGW | HLA−B*58:01 | Water soluble | Non−allergen | −0.86 | 0.4726 |
| ATGADHTVQL | HLA−A*68:02 | | | −0.84 | 0.4867 |
| ERIIDIRDSR | HLA−A*68:01 | | | −1.24 | 1.166 |
| FLGPSNHSL | HLA−A*02:01 | | | −0.85 | 0.4947 |
| FTSKIKARL | HLA−A*68:02 | | | −1.17 | 1.1466 |
| GSSRIGISY | HLA−A*30:02 | | | −0.88 | 0.5989 |
| ILYTRNYYK | HLA−A*03:01 | | | −0.18 | 0.6257 |
| ISYGRKKRK | HLA−A*30:01 | | | −0.97 | 2.3786 |
| KSYLDCQSW | HLA−B*57:01 | | | −0.77 | 0.6824 |
| LFTSKIKAR | HLA−A*33:01 | | | −1.05 | 1.1792 |
| LLSSAPLLFK | HLA−A*03:01 | | | −0.42 | 0.431 |
| LYTRNYYKI | HLA−A*24:02 | | | −0.18 | 0.4676 |
| MYSPRQANF | HLA−A*24:02 | | | −1.21 | 1.3485 |
| QSIPLLMHY | HLA−B*15:01 | | | −0.81 | 0.7776 |
| RERQEPPHKI | HLA−B*44:02 | | | −1.20 | 0.8659 |
| RGSSRIGISY | HLA−A*30:02 | | | −0.87 | 0.8534 |
| RIIDIRDSR | HLA−A*31:01 | | | −0.80 | 1.5405 |

also assessed the potential antigenicity of the alternative adjuvant. The vaccine sequence, comprising 781 amino acids, included the selected epitopes. Figs 3A and 3B illustrate the systematic scheme of the vaccine and the three−dimensional structure of the HIV vaccine.

**Table 4. A proportion of the population possesses HLA−binding alleles that can recognize and bind to epitopes. The evaluation of selected epitopes is conducted on a worldwide level, considering the average proportion of the population covered.**

| MHC class | Coverage | Population: |
|---|---|---|
| Combined | 80.31% | World |
| Combined | 93.49% | Japan |
| Combined | 90.87% | Korea; South |
| Combined | 88.57% | China |
| Combined | 90.69% | Hong Kong |
| Combined | 81.42% | Pakistan |
| Combined | 87.0% | Singapore |
| Combined | 81.2% | Philippines |
| Combined | 86.77% | Thailand |
| Combined | 95.69% | Taiwan |

## 3.7 Physiochemical properties of vaccine construct

The Allertop server was utilized to determine whether the immunizations possess allergenic properties, thereby establishing any association between the vaccinations and allergic reactions. Furthermore, the antigenic potential was assessed using the VaxiJen server, which produced a result of 0.782615, exceeding the threshold level of 0.4. The physicochemical characteristics of the vaccine construct were described using the ProtParam server, revealing a molecular weight of 87,199.65 daltons, a total of 781 amino acids, and an isoelectric point (pI) value of 10.12. Among the amino acids, 144 exhibited positive charges, while 68 displayed negative charges. Additionally, the half–life of the immunizations was determined to be thirty hours in vitro for human reticulocytes, whereas the half–life in vivo for yeast cells exceeded twenty hours, indicating a high degree of stability when administered to yeast cells. The instability index calculated a value of 31.76, suggesting that the protein is highly stable. Given the aliphatic index of 67.64, it can be concluded that the vaccine is extremely thermostable. The protein's hydrophilicity was evaluated, yielding a GRAVY coefficient of −0.686, which indicates that it is hydrophilic.

## 3.8 Modeling of secondary predication

We utilized the PSIPRED tool to predict the secondary structure of the final vaccine construct. According to the data obtained from the server, the vaccine structure comprises 31.48% alpha helices, 15.61% extended strands, and 52.91% coils. The secondary structure of the model is illustrated in Fig 4.

## 3.9 Tertiary structure modeling

Using the i−TASSER web server, we obtained the tertiary structure of the HIV vaccine construct. With the assistance of Galaxy refinement for online optimization, this structure underwent further refinement. Fig 5C illustrates the spatial organization of the HIV vaccine design.

## 3.10 Tertiary structure refinement

Through the utilization of the Galaxy web server (https://galaxy.seoklab.org/), the structural model, which was downloaded from the I−TASSER web server, was improved. These refinements aimed to minimize the energy associated with the molecular bonds. According to the

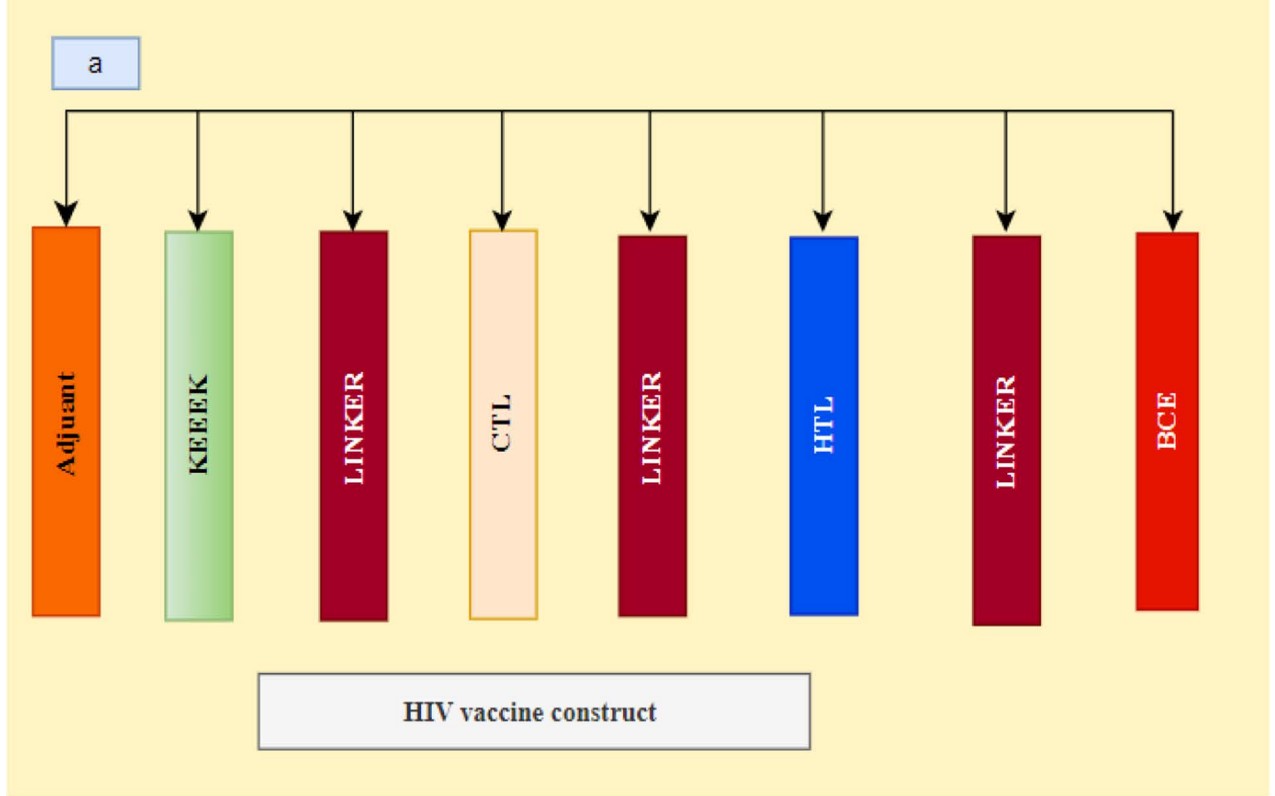

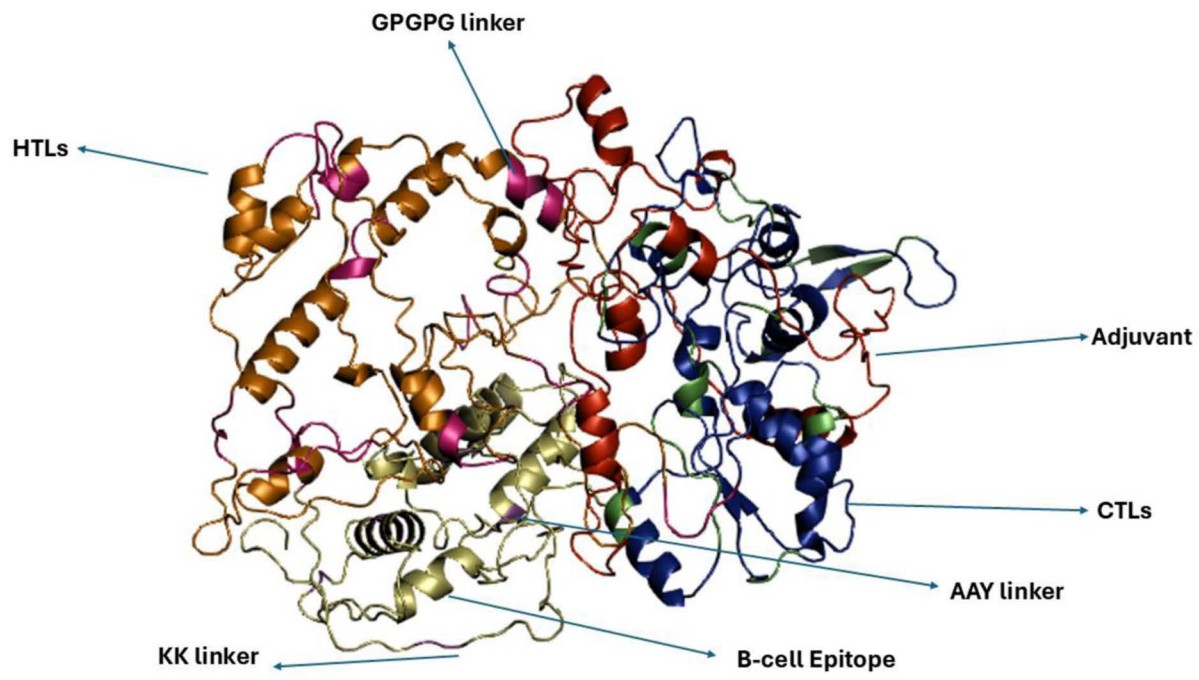

**Fig 3. A and B represents the epitopes of CTL, HTL, and CTL via a linker.** The tertiary structure of HIV vaccine.

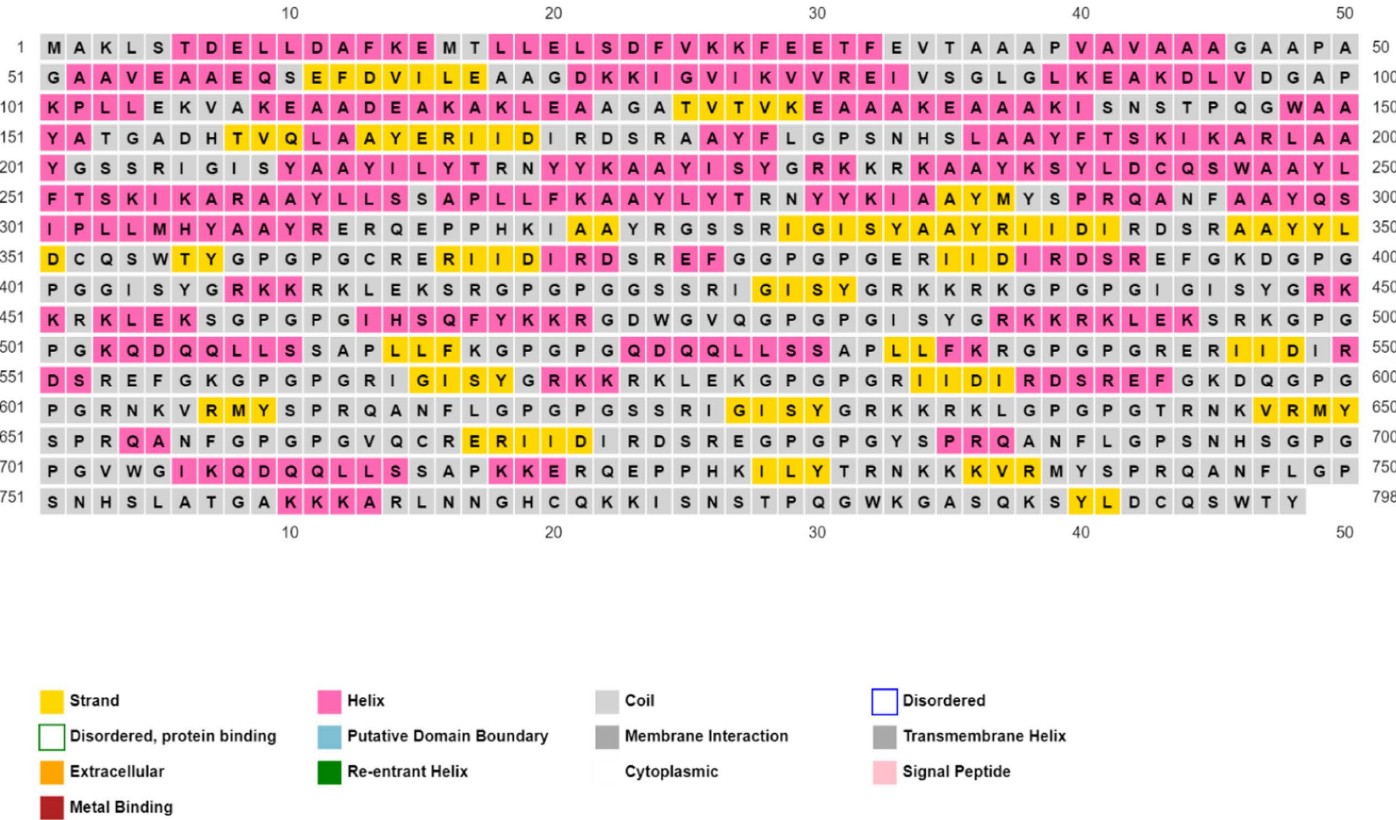

**Fig 4. The vaccine design exhibits a secondary structure comprising of several epitopes, which include alpha helices (31.48%), beta strands (15.61%), and coils (52.91%).**

predictions generated by the Galaxy web server, the Gag–Pol–vpr–Tat–Nef vaccine construct is anticipated to achieve an overall quality factor of 90.6%. The accuracy of the three–dimensional protein models intended for vaccination was validated by analyzing the dispersion of the Ramachandran plot, employing homology modeling techniques. An examination of the enhanced protein model using the Ramachandran plot revealed that 83.5% of the residues are situated in highly favorable regions, 14.6% are found in acceptable areas, and only 1.6% are located in disallowed regions, as illustrated in Figs 5a, 5b, and 5c [2]. This score is higher than what is typically observed in natural proteins of similar sizes

### 3.11 Disulfide bond

It was determined that the disulfide bonds were successfully identified by the Design v2.12 web server. A total of 32 pairs of amino acid residues were deemed suitable for generating disulfide mutations during the disulfide engineering process. Both the χ3 angle and the energy values of these pairings were analyzed with meticulous attention to detail. Following a comprehensive investigation, it was found that only four distinct combinations of amino acids met the specified criteria. These criteria included an energy score lower than 2.2 kcal/mol and a χ3 angle within the range of −87 to +97 degrees. Consequently, 10 different pairs of residues were modified: 6 THR – A 642 GLY, 7 ASP – 643 THR, 9 LEU – 12 ALA, 11 ASP – 375 GLU, 53 ALA – 239 SER, 143 SER – A 276 TYR, 555 PHE – 560 GLY, 557 LYS – 562 GLY, 751 SER – 775 SER, and 794 GLN – 797 THR. Figs 6A and 6B represent the submitted structure of the vaccine and the mutant structure.

B

A

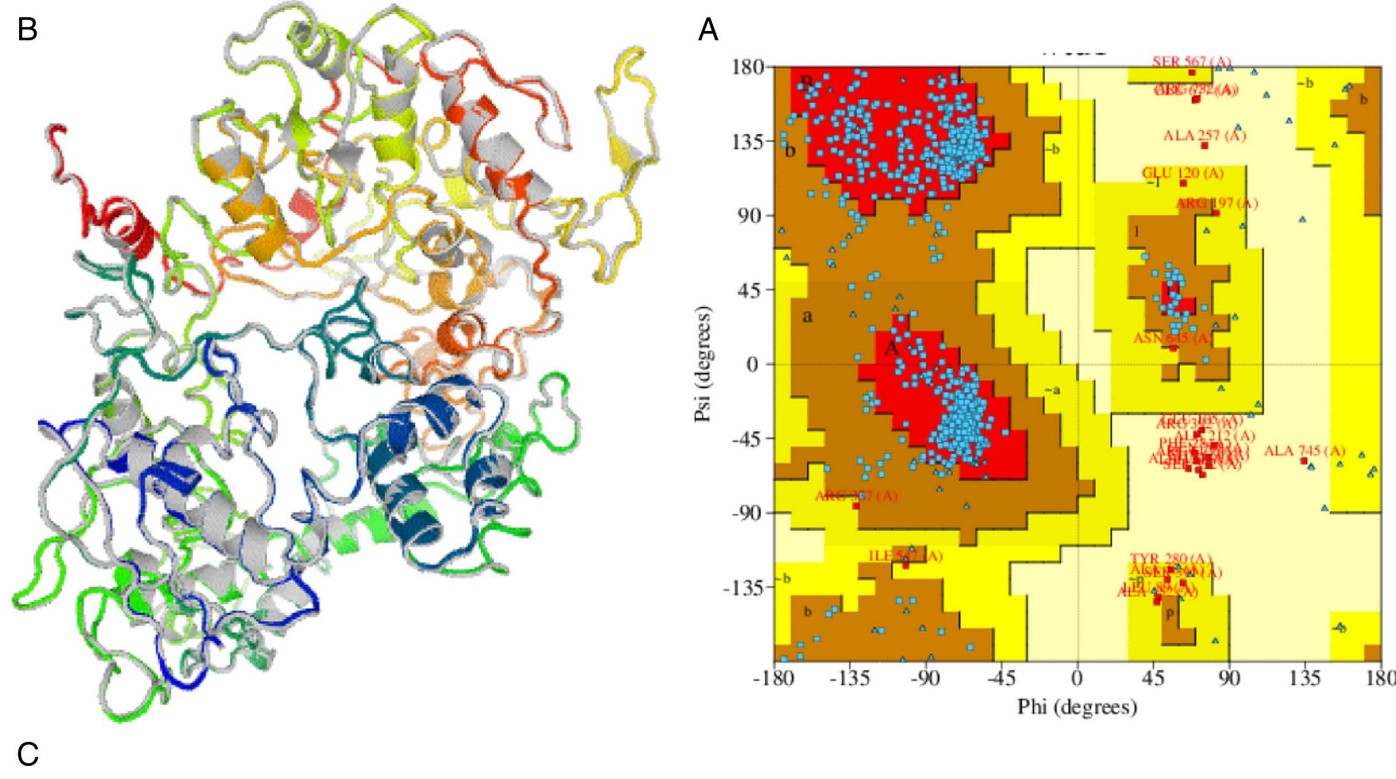

C

## PROCHECK statistics

### 1. Ramachandran Plot statistics

|  |  | No. of residues | %-tage |
|---|---|---|---|
| Most favoured regions | [A,B,L] | 536 | 83.5%* |
| Additional allowed regions | [a,b,l,p] | 81 | 12.6% |
| Generously allowed regions | [~a,~b,~l,~p] | 11 | 1.7% |
| Disallowed regions | [XX] | 14 | 2.2%* |
| Non-glycine and non-proline residues |  | 642 | 100.0% |
| End-residues (excl. Gly and Pro) |  | 2 |  |
| Glycine residues |  | 95 |  |
| Proline residues |  | 59 |  |
| Total number of residues |  | 798 |  |

**Fig 5.  (a,b,c) The Ramachandran analysis has improved the structure of the model, revealing that 83.8% of residues are in the most preferred zone, 12.3% are in the permitted region, and 2.3% are in the forbidden region. Additionally, 1.6% of residues are found in regions that are completely banned figure (a,b). Construction and verification of the ultimate subunit vaccination prototype. (c) The figure displays the ultimate 3D arrangement of the multi–epitope sub-unit vaccine obtained via the process of homology modeling and refining.**

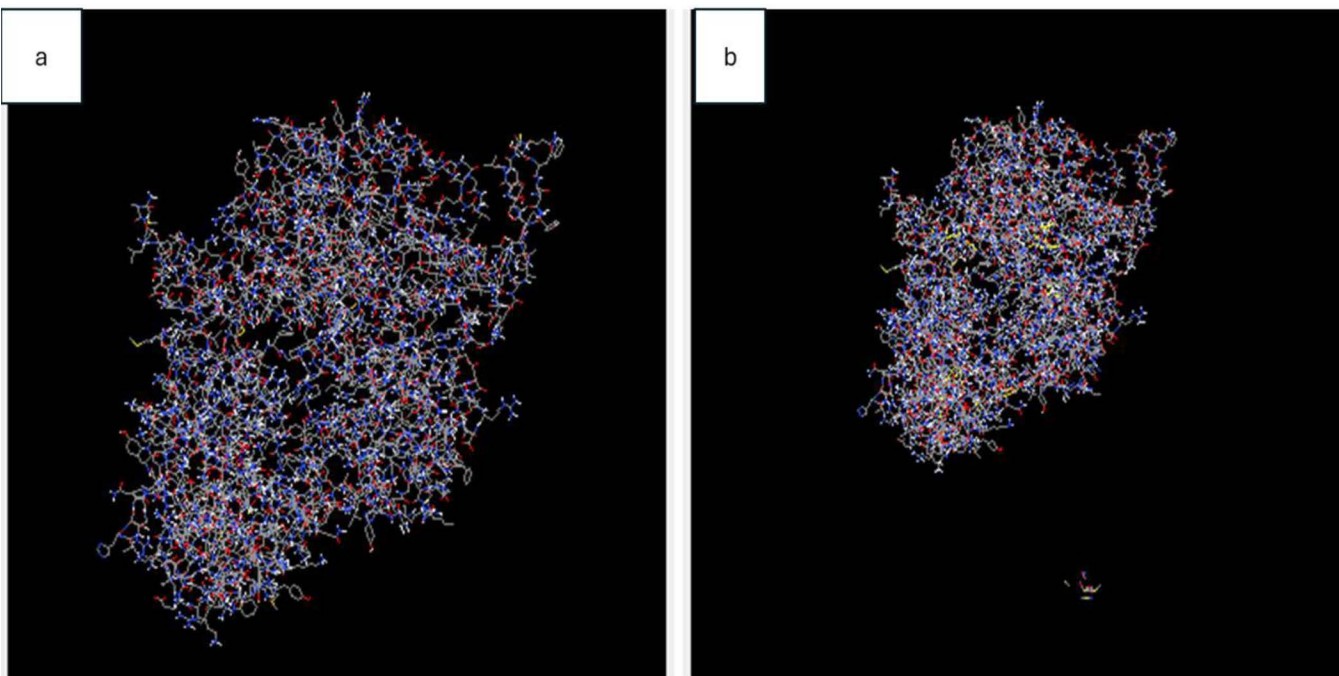

**Fig 6. Represents the disulfide bond in the 6(a) original vaccine construct and 6(b) mutation induce model by Design v2.12 web server.**

### 3.12 TLR docking

TLR receptors are a crucial part of innate immunity to recognize pathogens. These are used as primary targets in−silico vaccine design to enhance the vaccine efficacy by triggering an immune response. Hdock was utilized to connect Toll−like receptors (TLRs) with complexes of multiepitope peptides for protein−protein docking. Ten models were constructed for each docking scenario. Our selection process was optimized to include only those models that achieved receptor occupancy with the lowest energy scores. Our HIV vaccine candidate can bind to TLR3, TLR5, TLR7, TLR8, and TLR9 but they show strong affinity to the TLR3 and TLR5. During docking with TLR−3 and TLR−5, the Gag−Pol−Vpr−Tat−Nef vaccine construct achieved minimum energy levels of −336.17 and −376.66, respectively. The confidence scores for TLR−3 and TLR−8 are 0.9894 and 0.9764, as shown in Figs 7A and 7B. Fig 7B illustrates all the interacting residues between the TLR−3 receptor and the vaccine construct.

### 3.13 Prediction of the 3D structure of T−cell epitopes and molecular docking with HLA

The structure−based molecular docking was performed between the Helper T−cell epitopes and MHC−ii HLAs to determine the docking and confidence score by online HDOCK server. Fig 8(a, b) illustrates the docking between Helper T−cells (HTLs) epitopes, (ERIIDIRDSREF-GKD, CRERIIDIRDSREFG) and MHC−II HLA−DRB1*04:05 with binding−757.24, −229.92 with confidence score 1.00001 and 0.8318. similar in Fig 8(c, d, e): docking between HTLs epitopes (GISYGRKKRKLEKSR, ISYGRKKRKLEKSRK, RIGISYGRKKRKLEK) with HLA−DRB1*13:02 having docking score −756.24, −840.70, −801.34 with confidence score 1.0000, each HTLs epitopes. 8(f, g). depict the interaction between the HLA−DRB1*15:01 receptor and the Helper T−cell epitope (TRNKVRMYSPRQANF, RNKVRMYSPRQANFL). The

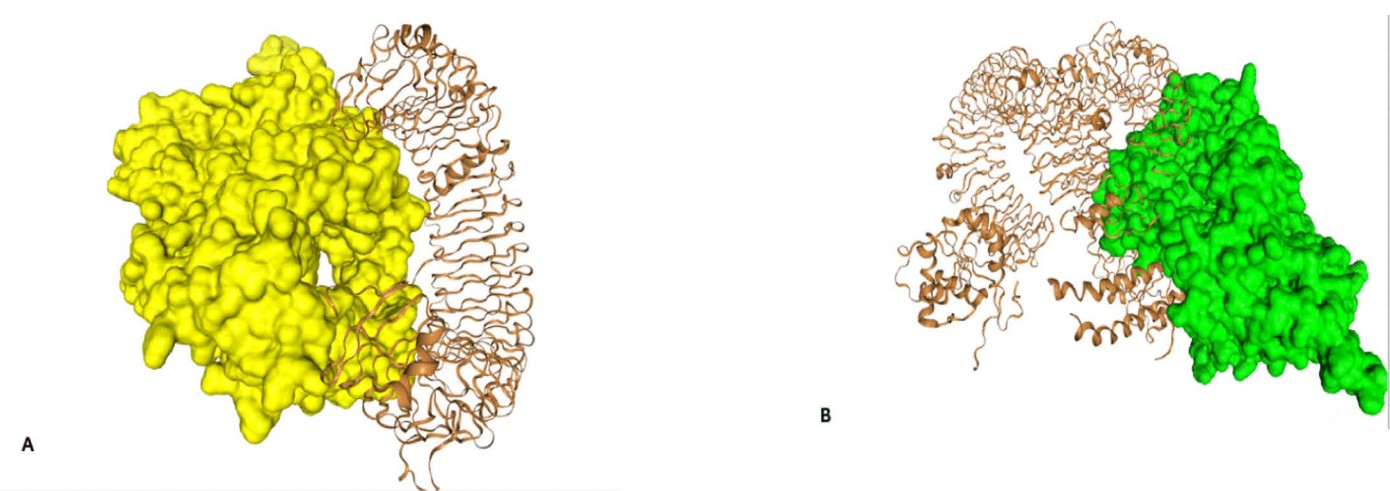

**Fig 7. The protein–protein docking research was conducted using Hdock to examine the interactions between the vaccine protein and TLR−3 in Fig 7(A).** The vaccine protein is represented in yellow when complexed with TLR−3. The F7 (b) represented all the interacting residues of TLR3 and the vaccine construct.

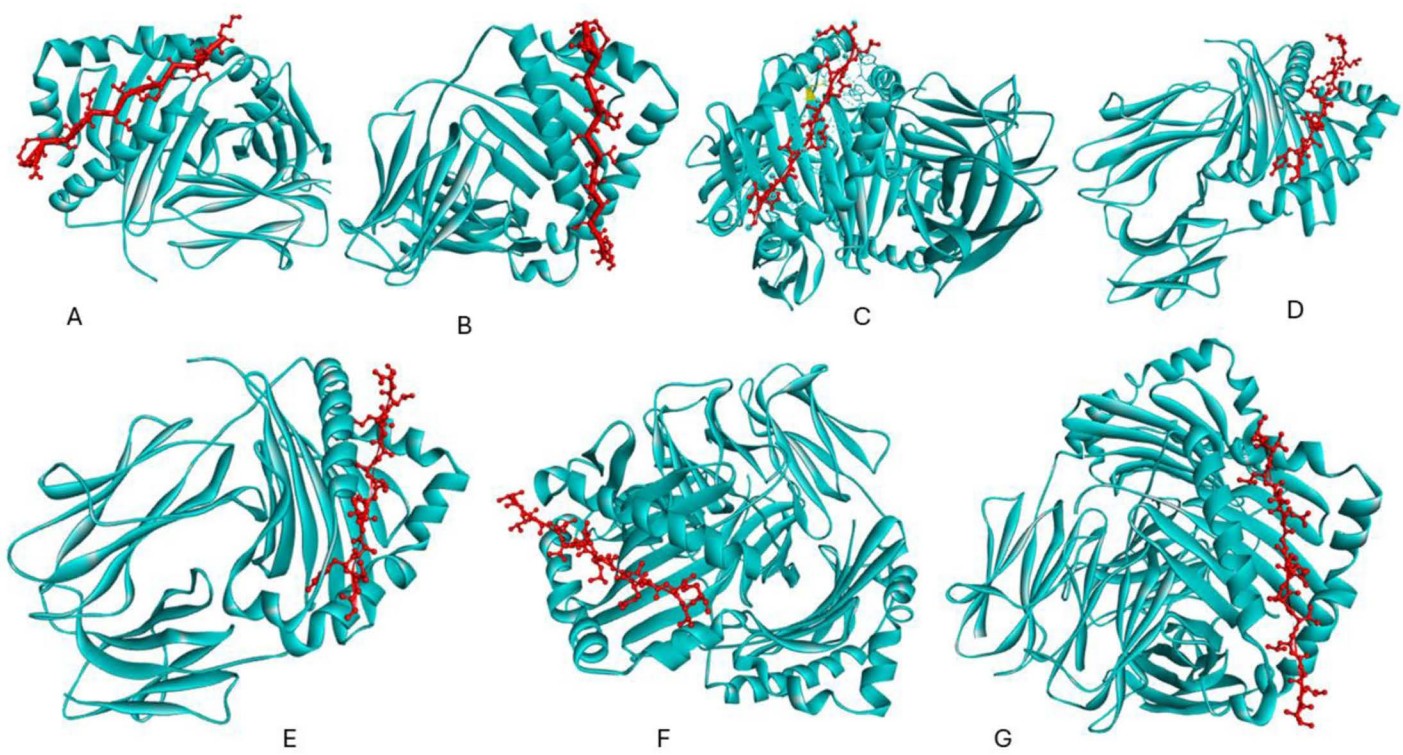

**Fig 8. Represents the molecular docking between helper T−cell epitopes and MHC−II molecules.** The helper T−cell epitope is represented in red colour while the MHC alleles are represented in cyan color. 8(a, b) represent the docking between T−cell epitopes (ERIIDIRDSREFGKD, CRERIIDIRDSREFG) with HLA−DRB1＊04:05 (c, d, e) docking between T−cell epitopes (GISYGRKKRKLEKSR, ISYGRKKRKLEKSRK, and RIGISYGRKKRKLEK with HLA−DRB1＊13:02. 8(f, g). represent the docking of T−cell epitope (TRNKVRMYSPRQANF, RNKVRMYSPRQANFL) with HLA−DRB1＊15:01.

confidence scores were −717.40 and −721.79, and the confidence score was 1.0000 for T−cell epitopes with HLAs.

An illustration of the molecular docking of the cytotoxic T−cell epitopes (CTLs), FLG-PSNHSL with the MHC−I allele HLA−A∗02:01 may be seen in Fig 9(a) with the docking score −523.75 and confidence score −0.9994. Fig 9b illustrates the docking that takes place between QSIPLLMHY and HLA−B∗15:01 with docking and confidence scores of −595.01 and.0999. Fig 9(c,d) illustrates the molecular docking that occurs between LYTRNYYKI and MYSPRQANF when HLA−A∗24:02 with confidence scores −605.10, −506.29, and 0.999, 0.9992. Similar to, the docking between ILYTRNYYK and HLA−A∗03:01 (−573.98, 0.9998), KSYLDCQSW and HLA−B∗57:01 (−543.50, 0.996), and ISNSTPQGW and HLA−B∗58:01 (−501.03, 0.9991) is shown in Fig 9(e, f, g).

### 3.14 Coden optimization

There exist notable differences between the microbial expression systems of humans and Escherichia coli K12 strains, necessitating codon modification to align them with the expression system of the human host. Consequently, the vaccine protein transformed reverse transcription to conform to the characteristics of E. coli K12. The optimized codon exhibited a codon adaptation index of 0.92, indicating a higher prevalence of frequently utilized codons. The GC content was used to examine the concentration of modified codons, yielding a result of 57.82%, which is considered favorable. Furthermore, it was observed that the restriction sites

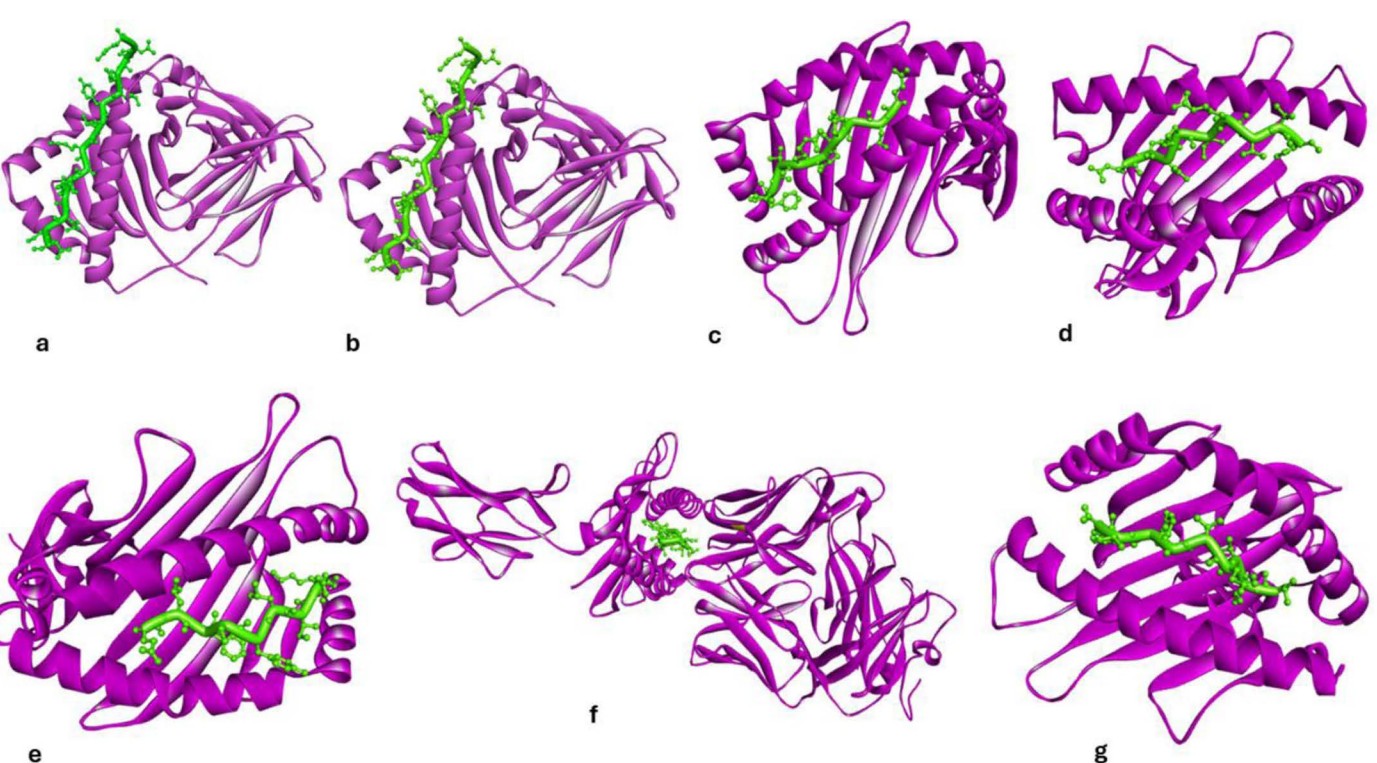

**Fig 9. Represents the molecular of cytotoxic T−cell epitopes with MHC−1 class HLA alleles.** Fig 9(a) represents the molecular docking of T cell epitope FLG-PSNHSL with allele HLA−A∗02:01. Fig 9b represents the docking between QSIPLLMHY and HLA−B∗15:01. The Fig 9(c,d) shows the molecular docking between LYTRNYYKI and MYSPRQANF with HLA−A∗24:02. Similarly, the Fig 9(e, f g) represents the docking between ILYTRNYYK and HLA−A∗03:01, KSYLDCQSW and HLA−B∗57:01 and ISNSTPQGW with HLA−B∗58:01.

BstZ17I and Eco53KI did not occur simultaneously. Therefore, this configuration was deemed suitable for cloning purposes. Subsequently, the modified codon, along with the restriction sites BstZ17I and Eco53KI, was incorporated into the pET28a(+) vector, resulting in the formation of a clone comprising 4950 base pairs. The intended sequence consists of 2378 base pairs, while the residing portion is attributed to the vector. We have used Pet28a for several reasons, such as its high expression of recombinant protein. It has a strong T7 promoter, which enables it to express proteins at a high level. It has a wide variety of restriction sites, which makes it an ideal candidate for cloning. pETa vectors can be engineered to include various fusion tags (e.g., His−tag, GST, MBP) that facilitate the purification and detection of the expressed proteins. The specific region within the pET28a(+) vector sequence is highlighted in purple. Fig 10 illustrates the structure of the vaccine in conjunction with the pET28a(+) vector.

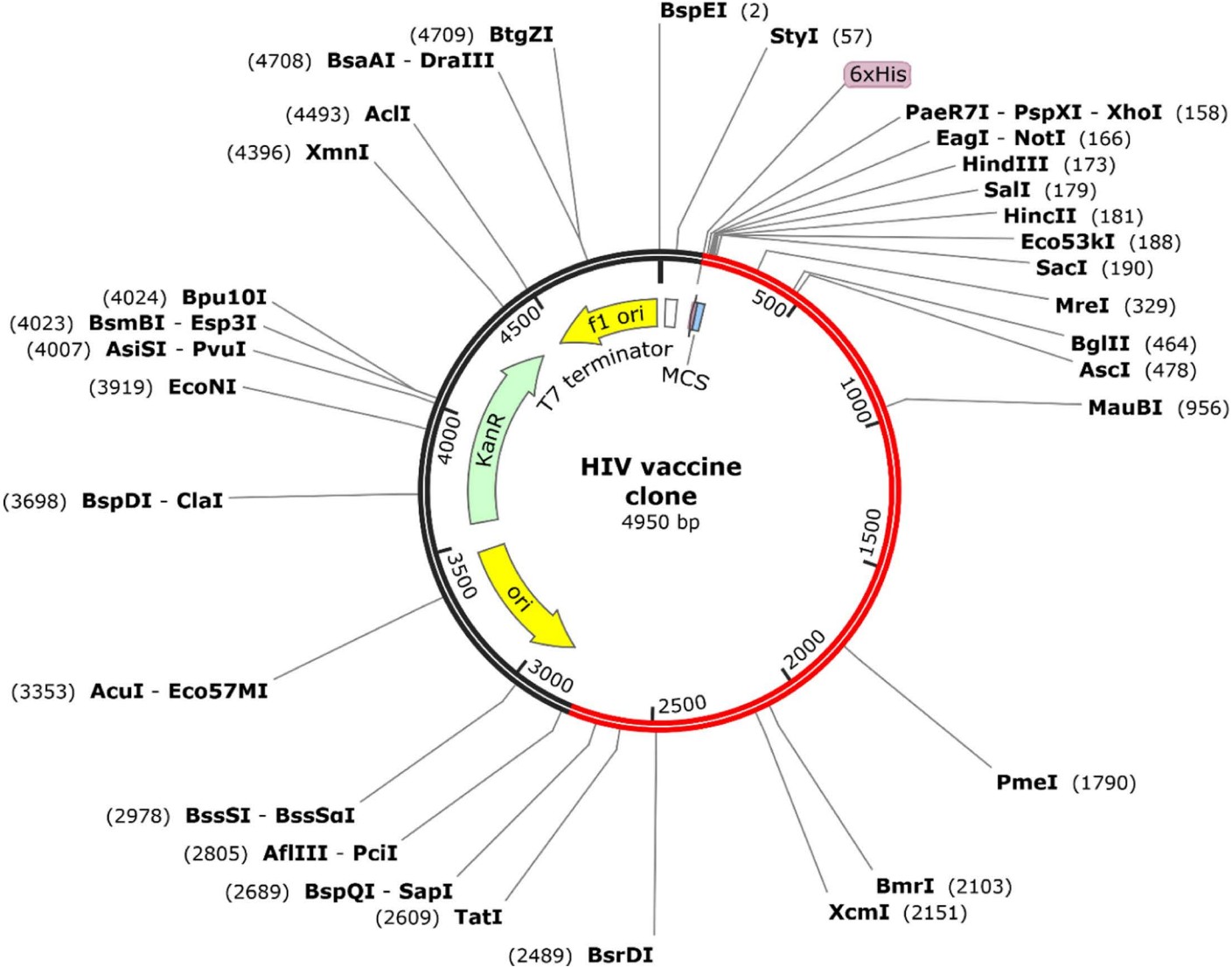

**Fig 10. The HIV construct is shown as a red line surrounded by a black circle, representing the in−silico process of cloning the HIV construct into the pET28a ( +) expression vector.**

## 3.15 MD simulation of HIV construct with TLR3 Complex

The purpose of the simulation study was to enhance our understanding of how atoms and molecules move within the structure of the vaccine. Fig 11A presents a representation of the peaks identified in the deformability graph, indicating significant deformability in specific regions of the protein that corresponds to these peaks. The B−factor graph, illustrated in Fig 11B, demonstrates the relationship between the docked complex from Normal Mode Analysis (NMA) and the Protein Data Bank (PDB), utilizing diagrammatic representations to clarify this association. Fig 11C graphically displays the eigenvalue of the complex, calculated to be $1.3467379e-09$, which reflects the eigenvalue at that specific moment. The covariance map of the complex, depicted in Fig 11D, visually represents the correlation between two residues by illustrating simultaneous motion. Correlated motion is shown in red, while distinct motion is represented in white. Blue indicates a motion that is separate from other movements and is not associated with their connections. Fig 11E presents an elastic map of the complex, where darker gray areas signify regions of higher stiffness, further emphasizing the relationships among the atoms. Finally, Fig 11F illustrates the variance graph, distinguishing between over-all variation, indicated in green, and individual variance, depicted in red. It is noteworthy that the total variation exceeds the individual variance.

## 3.16 Immune simulation

Using the C−IMMSIM interface, it was possible to successfully replicate the immunological reactions associated with HIV−1 vaccine designs. HIV−1 is known to elicit a robust immune response due to its ability to trigger a wide range of cytotoxic T−cell responses,

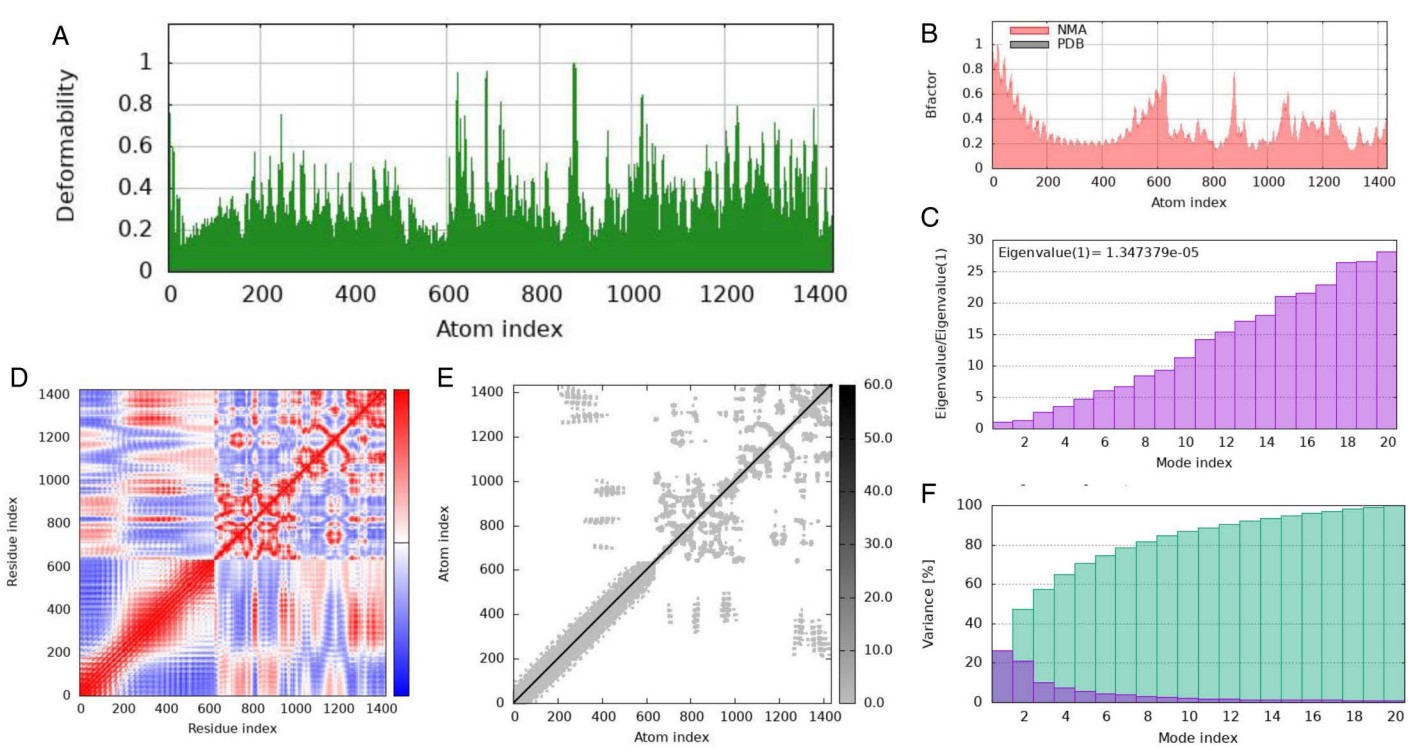

**Fig 11.** (a) Deformability, (b) B−factor, (c) Eigenvalues, (d) co−variance map (e) elastic network (darker gray regions indicate stiffer regions), (f) variance.

resulting in more effective immunological outcomes. Following the administration of the vaccine, multiple immunological responses were observed. Notable reactions included an increase in the number of helper B−cells and T−cells, enhanced production of IgG and IgM antibodies, and elevated levels of cytokines. HIV−1b has been shown to enhance antibody titers, specifically IgG2, the B isotype IgG2 cell population, the active TH cell population, the TH Mem (y2) cell population, and IL−2 levels. These observations support the findings. As illustrated in Figs 12A−12N, the immunological simulation generated data demonstrating that both recommended vaccines were capable of eliciting adequate innate and adaptive immune responses. These conclusions were confirmed following the simulation.

### 3.17  Molecular dynamic simulation of HIV vaccine construct and HIV construct with TLR complex

The Gromacs version 4.6.5, the molecular dynamics computation of the HIV vaccine was performed successfully.

**Root Mean Square Deviation (RMSD).**  The Root Mean Square Deviation (RMSD) is a standard measure of the distance of atoms or coordinates from a protein structure to those in reference as means for estimation of the overall stability and conformational changes [42]. As, in assessing the accuracy of structural alignments an RMSD value between 2 or 2.5Å is considered standard for structural comparisons [43]. The RMSD graph showed an initial increase reaching up to 2.5 Å as the protein relaxes from its starting conformation, followed by a stabilization phase till the end of the simulation. In the case of the protein under consideration, the presence of repetitive motifs (e.g., GISYGRKKRK) and flexible regions (e.g., Gly−rich) contributed to a higher RMSD compared to a more compact, globular protein as shown in Fig 13a. However, the stabilization phase suggests that the protein can adopt a relatively stable conformation despite its size and complexity.

**Root Mean Square Fluctuation (RMSF).**  During a molecular dynamic simulation, RMSF shows on average how much each of the residues in protein structure fluctuates. The high RMSF implies that these residues are likely to participate in the dynamic processes, including ligand binding or protein−protein interactions [44]. Corresponding to the flexible regions within the sequence of given protein, several peaks were observed. The 1st sharp peak was observed between 105−130 region contains a high proportion of polar or charged residues (E, K, T, and V). This is because such residues are typically solvent exposed and offer required flexibility. As the simulation progresses a sharp rise in RMSF was observed at 235−240 having A and Y (hydrophobic residues) whilst K and S constitute a polar sequence. It is worth noting that there is tyrosine (Y) which could engage in several interactions like hydrogen bonding and aromatic interaction. Also, the interactions of hydrophobic and polar residues could keep a dynamic nature. A wider fluctuation pattern was evident from 400−500 residues reaching up to 2.25 Å but as the simulation progresses, the protein gains a stable confirmation again in Fig 13b.

**Radius of gyration (Rg).**  The radius of gyration (Rg) measures the compaction and overall dimensions of a protein atrophy. It is the root mean square distance of atoms from the center of mass in the protein [45]. The Rg graph analysis and other analytical results obtained throughout the simulation revealed a correlation. Initially, the graph showed a sharp curvature, as the system rearranges structurally due to stabilization. This trend shows that the system undergoes dramatic conformational changes, gradually approaching a stable configuration indicating that the molecular system has attained a state of dynamic equilibrium, in which the overall structural integrity is preserved. Atomic position variations are minimized as shown in Fig 13c.

**Solvent Accessible Surface Area (SASA).** To better understand the folding/unfolding actions of the protein under consideration, we performed an analysis on SASA values during simulation. Commonly expressed in square angstroms, SASA analysis measured the surface area of biomolecules accessed by solvent ultimately leading to better understanding

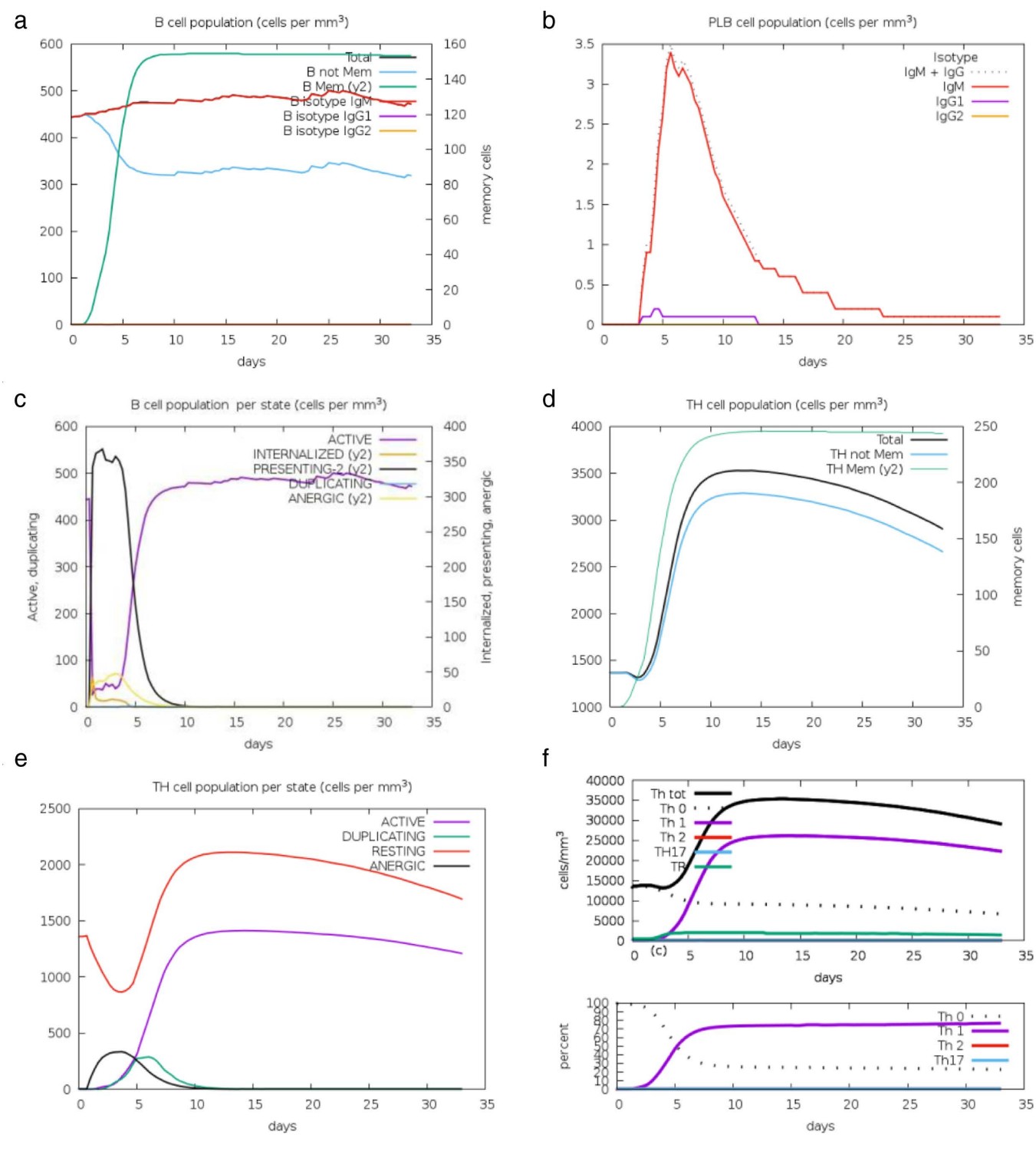

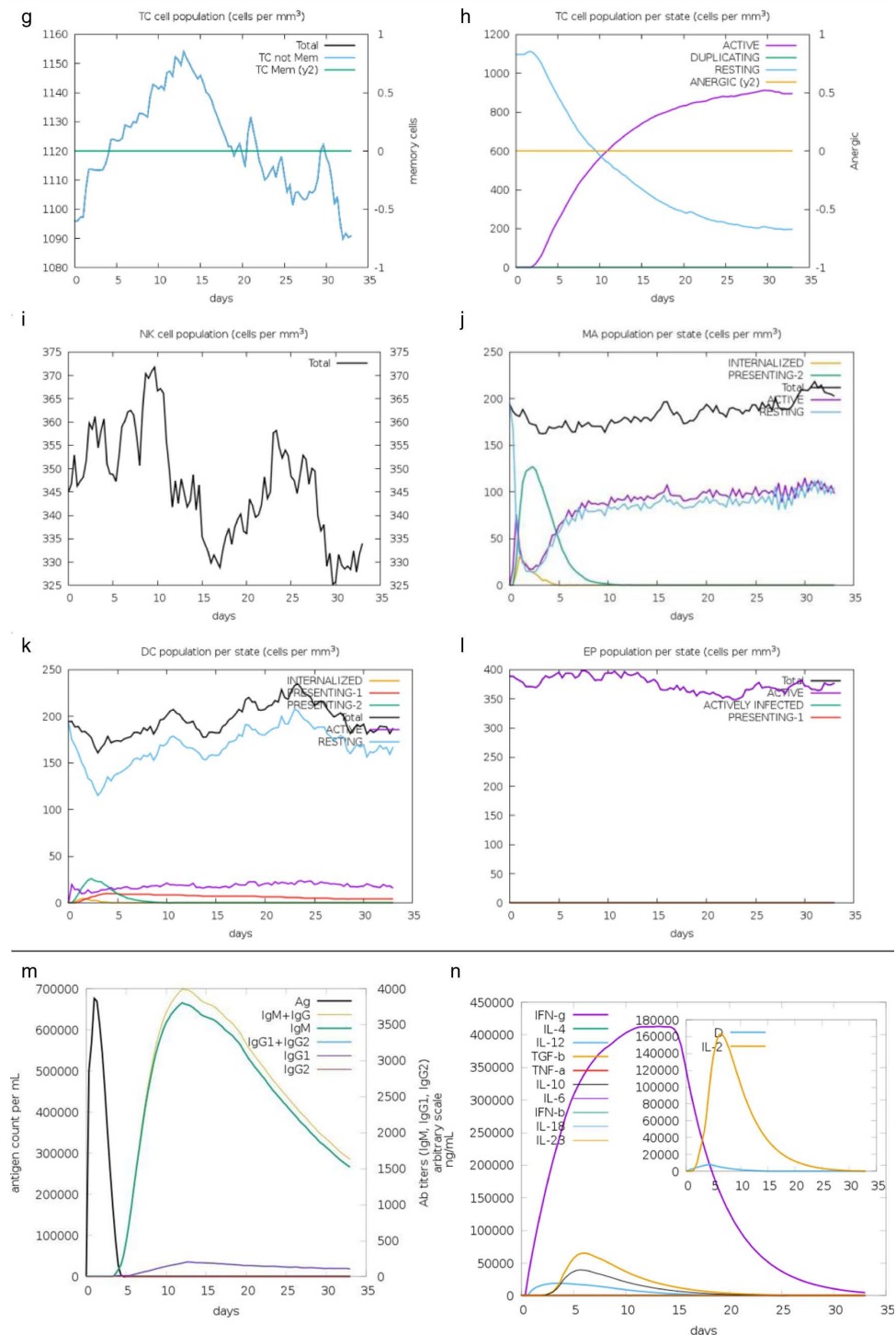

**Fig 12.** A−N The findings of the immunological simulation for the vaccine construct. The Graph indicates the strong response of the immune system by releasing antibodies (IgG and IgM) interleukins and cytokines.

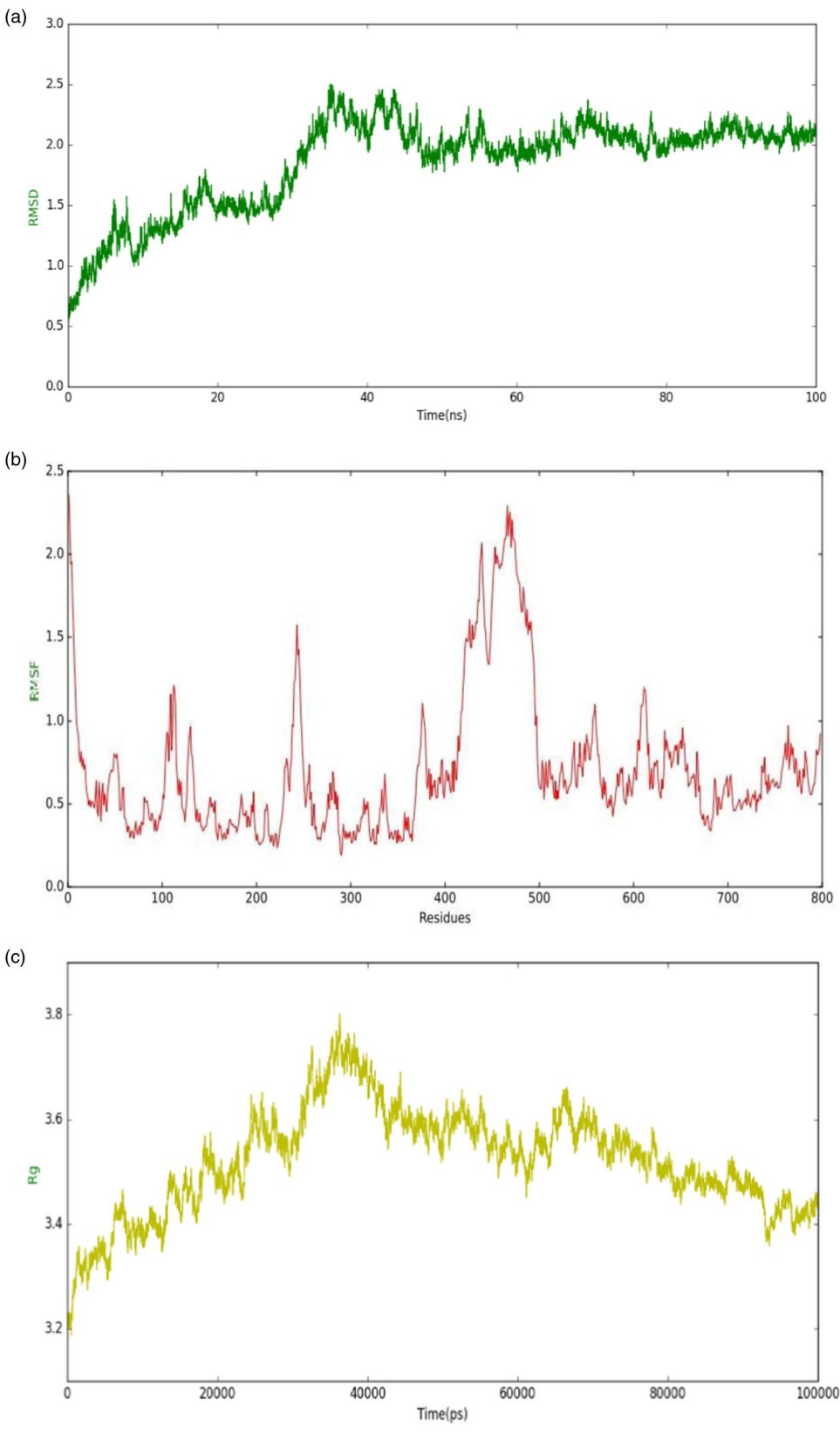

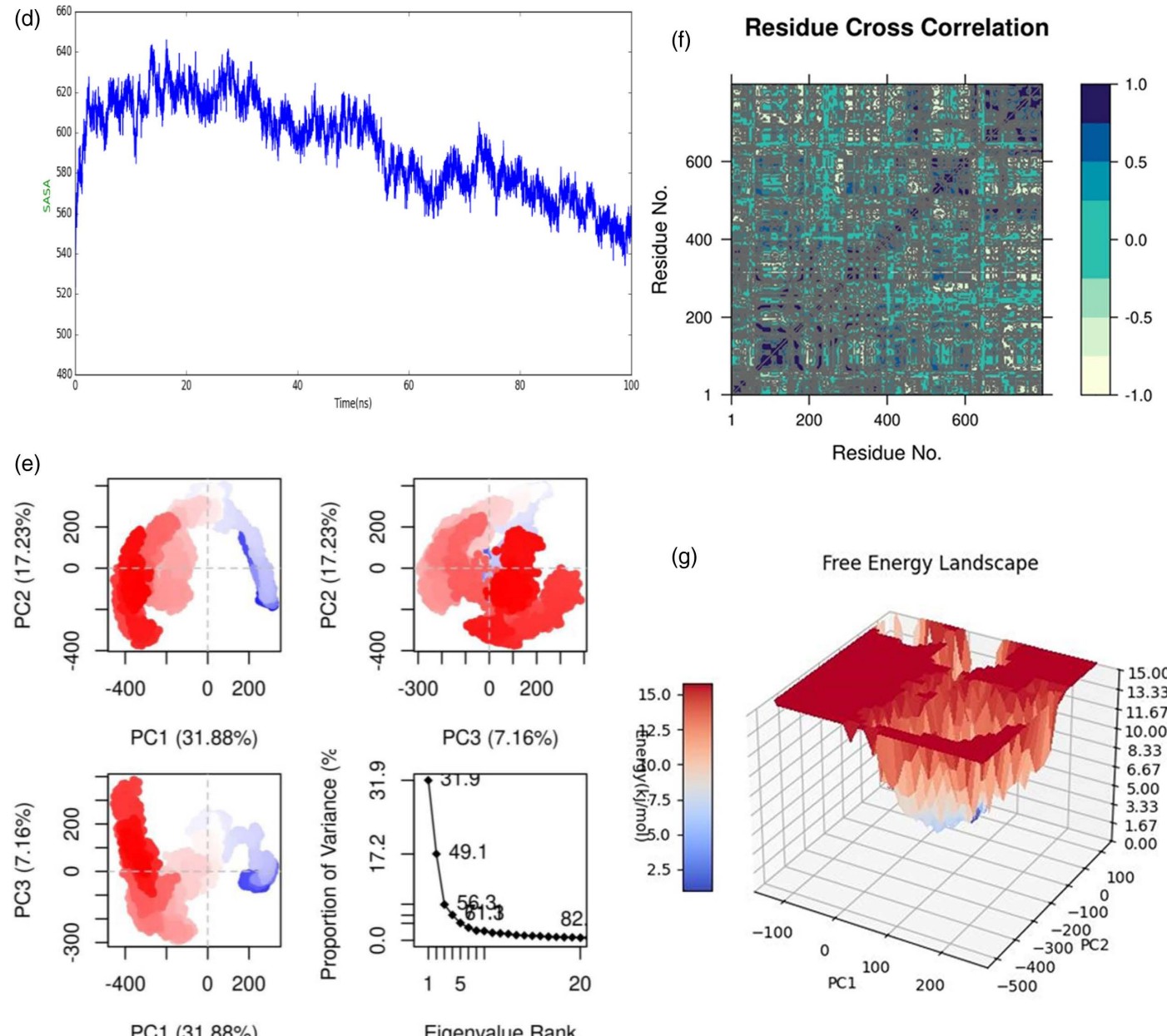

**Fig 13.** (a) shows the stable confirmation of the HIV vaccine construct with minor fluctuation at 40 ns and then again stable phase, (b) represents the RMSF plot having flexible and rigid regions. The peak shows the flexible region which are crucial for protein interaction, (c) represents the radius of gyration of the HIV vaccine construct which shows initial expansion followed by compaction and shows the stable conformation, (d) represents the SASA analysis of the HIV construct. The initial rise shows that the vaccine is hydrophilic due to polar amino acids while gradually decreasing in the peak shows the compactness of the HIV vaccine construct and the exposure of glycine−rich residues, (e) represents the PCA of the HIV vaccine construct, (f) represents the DCCM analysis of the vaccine of construct. The colors such as light green, green, and dark blue were used to analyze the results with dark blue indicating full correlation and colors close to light green in shade showing anti−correlation, (g) represents the energy state of the HIV vaccine. The overall state of the HIV vaccine revealed a stable conformation.

by exploring key biological functions, such as binding or enzymatic activity [44]. Fig 13d shows the SASA values that are plotted also showed a similar trend (i.e., progressive or rising pattern) to the Rg for all 100 ns trajectories. This is because of the given sequence, with more than 750 amino acids and many charged and polar residues (such as Lys, Arg, Asp, Glu) that

might affect its interaction with the solvent. During this exploration, potentially the protein exposes a greater surface area into solvent volume which corresponds to an increase in SASA value. However, the existence of multiple Gly− rich regions (i.e., the most flexible ones: GAPKPLLEKVAKEAADEAKAKLEAAGAT VTVKEAAAKE AA AK) might confer flexibility to the protein that enables versatility and allow the protein to switch to a more compact form, which resulted in a drop in SASA as the simulation progresses as shown in Fig 10d.

**Principal Component Analysis (PCA).** PCA was performed to explore the domain dynamics of protein by evaluating the concerted correlated motion denoted by the eigenvectors of covariance matrix across a 100ns MD simulation.

Eigenvalues typically indicate more coordinated conformational flexibilities concerning protein functions [46]. For the analysis, three conformations; PC1, PC2, and PC3 were employed. Every cluster had structural changes, according to the PCA analysis. The movements in the white zone were modest, those in the red region were the least flexible, and those in the blue region were the most significant. The protein's top 20 principal components (PCs) explained 88% of the total variation, as seen in Fig 13e. This suggests that the protein's phase space was more constrained, and its performance was less flexible. As indicated by the analysis, the protein accomplished a maximum variance of 31.88% for the 1st PC while for the 2nd and 3rd PC a variance of 17.23% and 7.16% were obtained respectively. The significantly pronounced movements for the 1st PC indicated the domain movements or the protein folding/unfolding transitions which are following the RMSF analysis. The overall values of variance revealed that the protein had a limited phase space and thus low conformational flexibility as shown in Fig 13e.

**Domain Cross−Correlation Matrix (DCCM).** Furthermore, the DCCM showed both outcomes − positive and negative for amino acid impacts. Residues are colored based on their sequence identity. Correlations near 1 represented residue shifting toward the same direction and correlations close to −1 could be interpreted as motions of single residues in opposite directions. Different colors mean different residue associations, e.g., dark color; high correlation; colors such as light green, green, and dark blue were used to analyze the results with dark blue indicating full correlation and colors close to light green in shade showing anti−correlation. As revealed by the plot, the domain (residue: 335−340) showed a positive correlation pattern while a negatively correlated motion was seen for the termini region of the protein. Fig 13f shows that overall, the high correlation in the protein was demonstrated by elucidating a DCCM plot, indicating that it had a more compact structure in form.

**Free energy landscape.** The confirmatory behavior of the HIV vaccine was further elucidated by constructing the Gibbs free energy landscape using the first two PCs. The varying conformational states of protein having lower energy were represented by a warm blue color. The FEL plot revealed that embedded within a single local basin, a single confined global energy minimum was evident. Overall, a stable state of protein was analyzed as shown in Fig 13g.

**MD simulation of Vaccine TLR3 complex**

**Vaccine−TLR3 complex MD simulations** To evaluate the stability of the vaccine−TLR3 complex, a 100 ns molecular dynamics [19] simulation was conducted using GROMACS, providing valuable insights into the stability and dynamics of this significant biomolecular interaction. The analysis involved multiple techniques, including root mean square deviation (RMSD), root mean square fluctuation (RMSF), radius of gyration (Rg), and solvent−accessible surface area (SASA), each contributing to a comprehensive understanding of the complex's behavior over time. The **RMSD analysis** revealed critical information regarding the system's stability. The observed plateau in the RMSD plot (Fig 14A) indicated that the system reached equilibrium, confirming that the simulation duration was adequate for capturing the

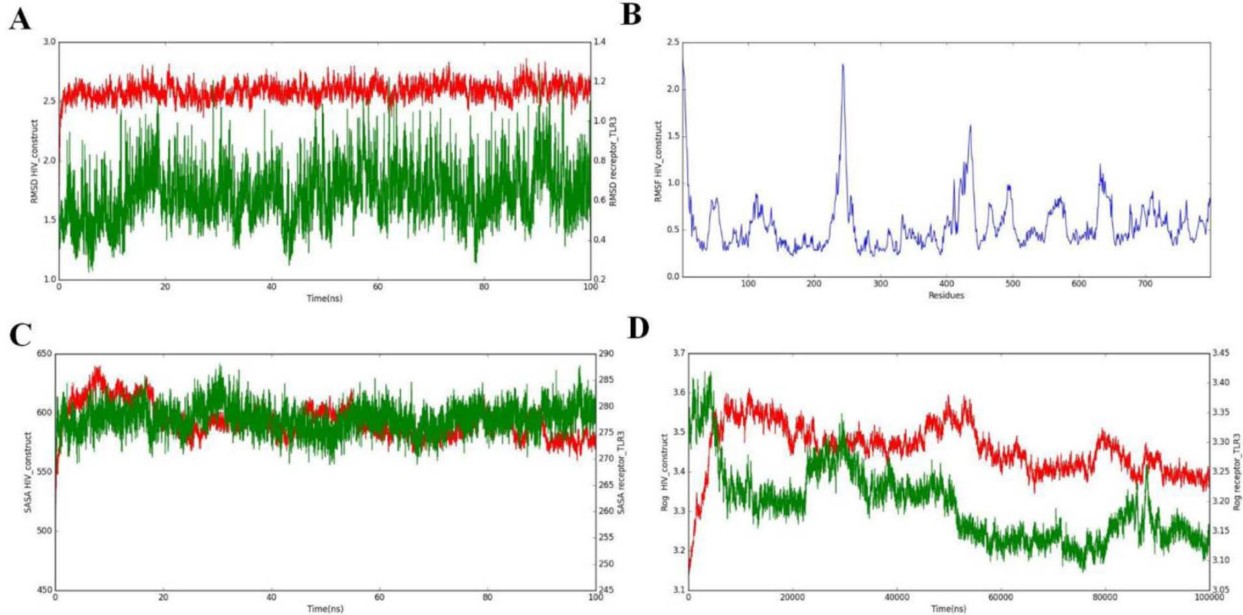

**Fig 14. The ligand–receptor complex (vaccine and TLRs) as simulated by molecular dynamics.** (14A) The docked complexes' stability throughout time is shown by the RMSD (Root Mean Square Deviation) study. (14B) The peaks in the RMSF (Root Mean Square Fluctuation) plot show areas of high flexibility. (14C) Information about the vaccine construct's surface exposure during the simulation is provided by the SASA (Solvent Accessible Surface Area) analysis. (14D) The vaccine design sustains a stable, compact shape during the simulation period, as shown by the Rg (Radius of Gyration) plot.

dynamics of the vaccine–TLR3 interaction. This stability is further supported by the absence of significant fluctuations, suggesting that the complex remains stable under dynamic conditions. Such findings are essential, as they imply that the vaccine maintains its structural integrity throughout the simulation, which is vital for effective immune engagement. In terms of **flexibility**, the RMSF analysis provided residue–specific fluctuation data, highlighting areas of higher mobility typically associated with flexible regions such as loops and coils in the ligand structure (Fig 14B). These fluctuations are indicative of regions that may adapt during binding interactions, allowing for optimal fit and enhanced binding affinity. Understanding these flexible regions can aid in designing more effective vaccines by targeting specific residues that contribute to receptor recognition. The **SASA analysis** tracked variations in surface exposure over time, showing a decrease in SASA (Fig 14C). This decrease suggests enhanced receptor–ligand interactions and indicates that as the simulation progressed, the complex adopted a more compact structure. This inverse relationship with RMSD indicates steady binding and structural stability between the multi–epitope vaccine and TLR3, reinforcing the idea that strong interactions lead to a more stable conformation conducive to immune activation. Additionally, the **Rg plot** (Fig 14D) highlighted the compactness of the protein complex, with an average radius of gyration of approximately 3.6 nm. This measurement reflects a tighter atom distribution around the center of mass, further supporting the notion that robust inter–component interactions are established throughout the simulation period. A compact structure is critical for maintaining functional integrity and enhancing receptor–ligand interactions. These results collectively demonstrate that the vaccine–TLR3 complex is structurally stable and compact throughout the 100 ns MD simulation. The robust inter–component interactions observed suggest a favorable conformation for effective immune response activation. These findings highlight not only the potential efficacy of multi–epitope vaccines but also

underscore the importance of molecular dynamics simulations in understanding biomolecular interactions. Future experimental validations will be essential to confirm these computational predictions and assess their implications for vaccine development against various pathogens.

**MMPBSA binding free energy analysis** The thermodynamic variable known as the free energy of binding ($\Delta$Gbind) is thought to be crucial for evaluating the favorable protein–protein interaction and its affinity for precise biological system modeling. In this context, the free binding energy of the MD simulations was determined using the g_mmpbsa tool. Solvent–accessible surface area (SASA) and unfavorable polar solvation energy (PSE), two of the previously mentioned favorable forces, are computed by MM/PBSA in Fig 15. The MM/PBSA calculated free energy of binding for the system was approximated as $-13950 \pm 15300$ kJ/mol. The docking was energetically possible, as demonstrated by the data's negative values of Gibbs free energy ($\Delta$G). As a potential vaccine candidate, the observed negative free binding energy value shows that the vaccination complex firmly attaches to the receptor as shown in Table 5.

**Discussion.** HIV, an epidemic virus, is responsible for the development of AIDS, a condition characterized by acquired immunodeficiency. Since its initial discovery, HIV has posed a significant threat to human health by specifically targeting T−helper cells. HIV is classified into two distinct groups, with HIV−1 being the predominant and primary cause of AIDS.

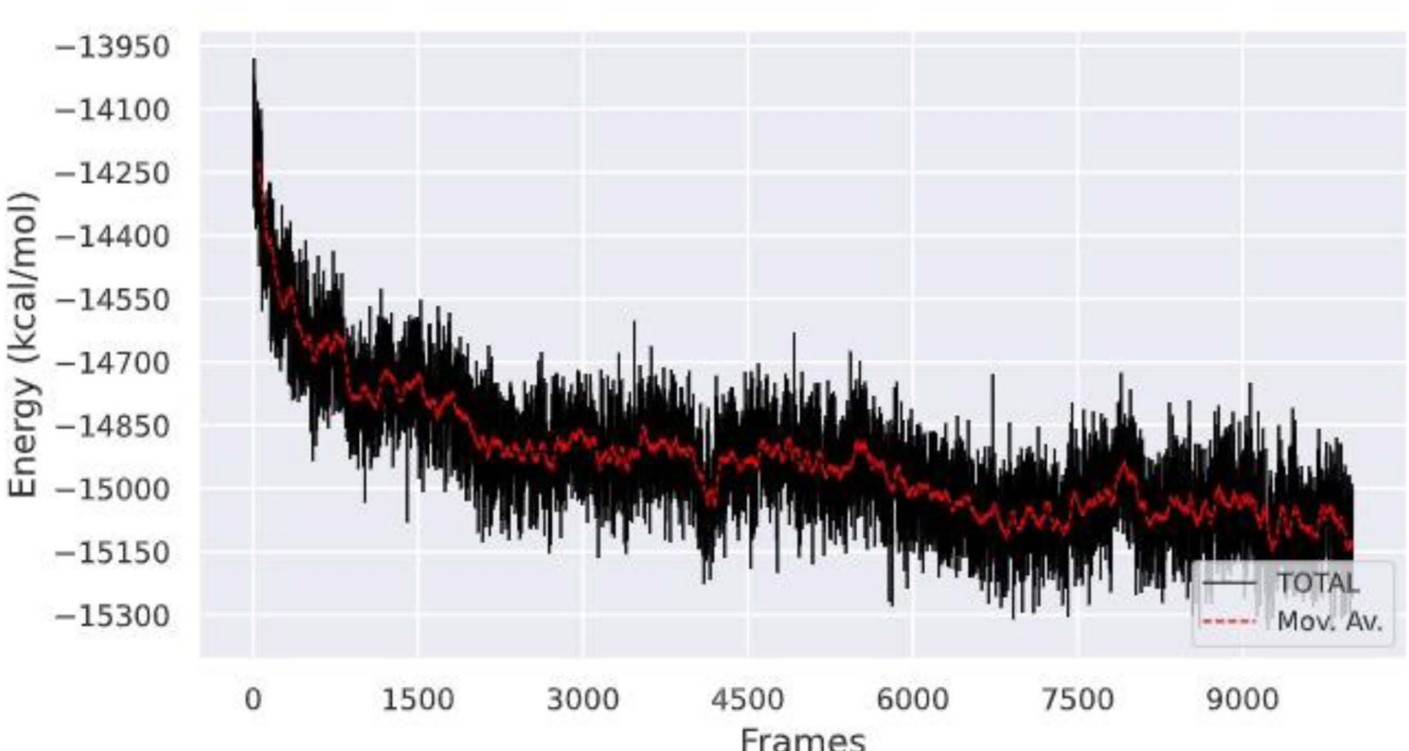

**Fig 15. MM/PBSA Analysis of Predicted Vaccine−Protein Interactions: Evaluating Binding Free Energies to Assess Stability and Efficacy of Designed Vaccine Candidates.**

**Table 5. represents the binding energy between HIV vaccine construct and TLR3 receptor.**

| BOND | 4659.808 |
|---|---|
| ANGLE | 11425.07 |
| DIHED | 14007.88 |
| UB | 1422.254 |
| MP | 735.1842 |
| CMAP | −1125.55 |
| VDWAALS | −8908.26 |
| EEL | −56559.1 |
| 1−4VDW | 3884.669 |
| 1−4 EEL | 50554.55 |
| EGB | −43461 |
| ESURF | 681.8187 |
| GGAS | 20096.54 |
| GSOLV | −42779.2 |
| TOTAL | −22682.6 |

Furthermore, despite the progress made in the treatment of AIDS with antiretroviral drugs, there is an urgent need to develop a robust vaccine against HIV−1 to effectively manage this devastating global epidemic. Research has demonstrated that most existing vaccines have not been successful to date. This is partly due to the fact that these vaccines are not particularly effective against the rapidly evolving virus. Previous vaccines were designed to target only a limited number of HIV strains, which contributes to their ineffectiveness [47]. As demonstrated by clinical studies utilizing the EP HIV−1090 vaccine, the ineffectiveness of HIV−1 vaccines may be attributed to their inability to elicit responses from both helper T−cells and cytotoxic T−cells. Additionally, the failure to develop broadly neutralizing antibodies, as evidenced by trials conducted on BALB/c mice using three different multi−epitope vaccines, further underscores the gravity of this issue. Collectively, these factors significantly contribute to the overall ineffectiveness of the vaccines. Due to the limited efficacy of immunizations against HIV−1, both of these elements have likely played a substantial role in the current situation [48,49]. It is also possible for immunizations to fail to elicit the appropriate cytokine response or to induce the necessary innate immune response. The methods employed in this study are comparable to those utilized in previous research aimed at developing multiepitope vaccines against various viruses, including HIV, Orthohantavirus, and T. whipplei [50]. Nevertheless, alternative multiepitope vaccinations demonstrate limited efficacy in eliciting an immune response against only a select few genotypes of the virus, which constitutes a significant limitation of these vaccines. Pandey and his colleagues have undertaken efforts to improve a vaccine that provides targeted protection against subtypes B and C of the group M virus, particularly concerning HIV. The current research includes a comprehensive analysis of the various potential HIV genotypes, which is a critical aspect of the study [51].

The integrase enzyme is the main focus for designing vaccines in the HIV−1 life cycle [52]. There are distinct and conserved sections of a protein sequence that have evolved independently throughout the protein's evolutionary history. These regions are referred to as protein domains. Not only do these domains reveal the structural characteristics of the entire protein, but they also provide insights into its functional properties. Consequently, these protected areas have the potential to serve as targets for the development of novel vaccines. In the quest to produce a vaccine against the highly infectious disease known as HIV, researchers

face significant challenges, primarily due to a lack of understanding regarding immunity to the virus. This knowledge gap is the main reason for their difficulties. In contemporary efforts to prevent infectious diseases, immunotherapy is widely regarded as the most effective and efficient approach available. During the vaccine development process, immunoinformatics—a subfield of bioinformatics—has been employed [53]. This approach has successfully achieved its objectives by enabling the timely and cost−effective analysis of immunological data [54]. Multiple research studies have been carried out to develop multi−epitope vaccines for various viruses, such as Dengue virus [40], Hepatitis B virus [55], Hepatitis C virus [56], Ebola virus [57], Chikungunya virus [58], Avian influenza A (H7N9) virus [59], Zika virus [60], Classical swine fever virus, Nipah virus, and Norovirus [19]. Olga, along with her research colleagues, investigated the Leishmania infantum eukaryotic initiation factor (LieIF) protein. The experimental work was primarily conducted by Olga. To facilitate the analysis of the protein, in silico methodologies, specifically antigenicity prediction algorithms, were employed. This approach enabled the identification of potential LieIF epitopes that may be presented by H−2 class I and II alleles. Consequently, the researchers prepared five LieIF peptides that could be beneficial for their ongoing research endeavors. The next phase involved evaluating these peptides in a controlled laboratory setting to ascertain their ability to activate an immune response. Furthermore, to investigate the significant effects of these peptides on the immune system, dendritic cells derived from bone marrow were utilized in the laboratory. This comprehensive approach was undertaken to gather relevant information regarding the immunological implications of the LieIF peptides [61]. Andres et al. employed immunoinformatics approaches in their study to identify highly conserved class I and II T−cell epitopes in seven common strains of the influenza A virus (IAV) found in pigs in the United States. This identification followed the researchers' comprehensive investigation. These specific regions of a protein, known as epitopes, are anticipated to bind to various swine leukocyte antigen (SLA) alleles that are available for commercial use. This prediction has been made based on their findings. A vaccine was developed using plasmid DNA containing multiple epitopes, which was subsequently tested on live pigs. Ultimately, the vaccination proved to be successful [62,63].

To make predictions about epitopes, the research utilized the complete HIV−1 genome. This was achieved by identifying conserved regions, including Gag, Pol, Vif, Vpr, Tat, and Nef. The presence of various component proteins and polyproteins is essential for the HIV−1 life cycle. These proteins are crucial for the virus's replication and pathogenicity. Consequently, focusing on these proteins may represent an effective strategy for preventing or treating HIV−1 infection. Gag, Pol, and Env are examples of polyproteins that have been identified as immunogenic, meaning they have the potential to elicit a strong immune response [15]. Moreover, HIV−1 exhibits significant genetic variability, leading to considerable differences in protein sequences among various viral strains. Consequently, by strategically targeting specific proteins, it is possible to develop a vaccine that effectively combats a broad spectrum of HIV−1 strains. Multi−epitope vaccines, which are designed to elicit a robust immune response, were created using a combination of 41 epitopes. Initially, the process of identifying B and T−cell epitopes was conducted, as this is a crucial first step in vaccine development. Subsequently, a filtering technique was employed to identify T−cell epitopes that overlapped with B−cell epitopes and stimulated a strong immune response [64] Therefore, the chosen epitopes possess the capability to elicit cellular T−cell, helper T−cell, B−cell, and IFN−ɣ responses. At present, there are ongoing clinical trials to examine subunit vaccinations that include cellular and helper T−cell epitopes [65]. The safety of these epitopes was assessed by toxicity and allergenicity studies. Further the MD simulation predict the stable confirmation of HIV vaccine.

In silico vaccine design to be useful for human use, our HIV vaccine must go through several phases of the systematic, regulated preclinical and clinical trial process before evolving to a ready for−market vaccine.

**Preclinical phase.** This first phase of our HIV vaccine, is the preclinical phase, entails the use of laboratory and animal studies to evaluate the safety and immunogenicity of the vaccine candidate. Initially, the vaccine construct is to be incorporated into the right expression vector, for example pET28a vector, and the expression host may be bacterial or mammalian cells [66]. The purified protein can even be further tested in vitro by exposing the human or animal immune cells and measuring their immune activities, such as the activity of cytokines or T−cells. Animal studies are the next level where immunogenicity and some levels of safety are measured in a relevant animal model such as mice and monkeys [67].

**Preparing for clinical trials.** After the positive outcome of the preclinical studies the next step is Clinical trial preparation which includes the filing of an Investigational New Drug (IND). All preclinical information, including immunogenicity, toxicity, protection data, vaccine manufacturing procedures, product purity, and stability are contained in this application [68]. All vaccines must meet GMP set by the regulatory authorities including the FDA for the USA or EMA for Europe to guarantee that the vaccine is produced conforming to high standards. Thus, after the clinical trials approval, Phases I testing regarding the safety in a limited number of healthy people can be conducted in parallel with further Phases. Phase II trials involve a larger group to determine dose−response/repetition regime and immunogenicity profiles with a safety checker. Phase III trials are to test the vaccine's effectiveness in thousands of people to ensure that the vaccine works for all and to find rare side effects [69].

**Significance.** The importance of the transfer of an in silico−design vaccine through preclinical and clinical trial phases is due to the prospects for fast and specific development of vaccines. Computation in design is vast and efficient for modeling immune responses; one can predict antigens, the stability of the proteins, and epitope regions before conducting lab testing [70]. Moreover, by using in silico approaches, one can define specific monopathy that a vaccine should stimulate, which theoretically makes the vaccine more effective and safer [71]. When refined and experimented with, a computationally designed vaccine is quite flexible and easy to tweak to face emerging subtypes or other related diseases, making it improvable in responding to different infectious threats [72].

**Limitations.** Nevertheless, there are some drawbacks to the in−silco approach to designing vaccines. Predictive models remain equally sensitive to the quality of the input data and can only partially mimic in vivo immune responses [73]. When using computational methods in the solution−finding process, there are physiological and atomic cellular level interactions that are not modeled to the letter and thus variations can be observed between the modeled outcome as opposed to the actual outcome. Thus, there is a wide gap of 10−15 years of development and careful preclinical and clinical development to check immunogenicity, safety, and efficacy. Further, some of the components of the vaccine constructs, which seem to be efficacious while modeled computationally may not fluoresce in protein expression, stability, or immunogenicity when translated into real biological systems. In conclusion, in silico design show great promise in revolutionizing vaccine development but needs proper incorporation with experimental work to accomplish some of its drawbacks and achieve favorable clinical results [16].

**Conclusion.** The global AIDS epidemic continues to pose a significant public health challenge, with the most promising intervention being the development of an effective and reliable vaccine. This is particularly noteworthy given the substantial advancements in medical science observed in the 21st century. Current immunization trials have yielded

unsatisfactory results, likely due to their inability to elicit robust responses from the cellular, humoral, and innate immune systems. In light of these challenges, the results of this investigation support ongoing efforts to create an HIV−1 vaccine. By identifying conserved regions of the HIV genome, we successfully retrieved the complete genome. Subsequently, we pinpointed the most immunogenic sites suitable for B cell lymphocytes (BCL), cytotoxic T lymphocytes (CTL), and helper T lymphocytes (HTL) within the human body. These regions, along with the necessary adjuvants and linkers, were utilized to achieve the objective of constructing a subunit vaccine. Additionally, the vaccine underwent characterization to assess its allergenicity, antigenicity, physicochemical properties, in silico cloning, and disulfide engineering parameters. Structural analysis of the vaccine was conducted in both two−dimensional and three−dimensional formats. The subsequent phase involved molecular docking and dynamics analysis to examine the stability and strength of the interaction between the vaccine construct and the Toll−like receptor 3 (TLR−3) complex. To validate these findings, it is essential to conduct experimental verification of the vaccine protein, which represents a crucial step in the development process.

## Acknowledgment

"The authors extend their appreciation to the Deanship of Research and Graduate Studies at King Khalid University for funding this work through Large Research Project under grant number RGP2/499/45

## Author contributions

**Data curation:** Syeda Maryam Hussain.

**Formal analysis:** Akmal Zubair.

**Supervision:** Akmal Zubair, Muhammad Ali.

**Validation:** Ahmed Al−Emam.

**Visualization:** Ranya Mohammed Elmagzoub.

**Writing – review & editing:** Akmal Zubair.

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
