## [Decision Letter · Decision Letter 0]

30 Oct 2024

PONE-D-24-40708Targeting HIV-1 Conserved Regions: An Immunoinformatic Pathway to Vaccine Innovation for the AsiaPLOS ONE

Dear Dr. Zubair,

Thank you for submitting your manuscript to PLOS ONE. After careful consideration, we feel that it has merit but does not fully meet PLOS ONE’s publication criteria as it currently stands. Therefore, we invite you to submit a revised version of the manuscript that addresses the points raised during the review process. The reviewers have identified several major concerns regarding methods used and/or results presented in the manuscript. Please go through the comments carefully and ensure that all concerns are adequately addressed in the revised manuscript. 

We look forward to receiving your revised manuscript.

Kind regards,

Syed Hani Abidi

Academic Editor

PLOS ONE

3. In the online submission form, you indicated that [Data will provide upon the request].

Reviewers' comments:

Reviewer's Responses to Questions

**Comments to the Author**

1. Is the manuscript technically sound, and do the data support the conclusions?

Reviewer #1: Yes

Reviewer #2: Yes

2. Has the statistical analysis been performed appropriately and rigorously? 

Reviewer #1: N/A

Reviewer #2: N/A

3. Have the authors made all data underlying the findings in their manuscript fully available?

Reviewer #1: Yes

Reviewer #2: Yes

4. Is the manuscript presented in an intelligible fashion and written in standard English?

Reviewer #1: Yes

Reviewer #2: Yes

5. Review Comments to the Author

Reviewer #1: The authors have conducted the study ‘Targeting HIV-1 Conserved Regions: An Immunoinformatic Pathway to Vaccine Innovation for the Asia ” via comprehensive bioinformatics approaches, which makes for a valuable vaccine candidate. The results would introduce a vaccine candidate that could be taken to in vivo and possibly clinical studies. However, before considering it for publication, several points should be considered.

Major comments:

1. The abstract should be a little more detailed. It does not clearly describe steps that assess the immunogenic capability of the vaccine construct. In addition, there is no conclusion at the end of the Abstract.

2. In the last paragraph of the Introduction, there was no hypothesis for this study, the authors should clarify the aims and hypothesis of this study.

3. What’s the difference between section 2.11 (Simulation-based computational approach) and section 2.14 (immune simulations)?

4. There are several published papers (PMID: 29571972, PMID: 31606417, PMID: 39331650, PMID: 37226084) on multi-epitope designed against HIV, providing advantages of your proposed vaccine.

5. It is not clear why the authors selected TLR3 and TLR5for the docking of the vaccine construct. Please provide the rationale for using this receptor molecule.

6. The authors should provide the affinity between vaccine and TLR3 -5, which could show how the vaccine binds to TLR3 or TLR5. Perform the MD simulation with GROMACS trajectory and explore carefully to ensure the vaccine consistently binds to TLR3 or TLR5.

Reviewer #2: 1. Please perform the molecular docking between T-cell epitopes and MHC molecules.

2. Authors should evaluate the antigenicity, allergenicity and solubility of the vaccine construct.

3. In the manuscript, the authors should provide a logical reason for using this type of linkers in the structure of the vaccine by citing a valid reference.

4. Authors should predict discontinuous B-cell epitopes on the 3D structure of the vaccine.

5. The authors should present the results of the "Molecular Dynamic Simulation" and "Immune Simulation" sections in more detail.

6. The authors should further discuss why the current vaccine constructs is more advantageous than the previous ones. How do you see the future steps for your vaccine? What are your next planned steps? Add a discussion regarding the next needed steps, Limitation, and etc.?

7. Please provide the MM-PBSA calculations of the simulated trajectory.

8. Please explain why you selected the pET28a plasmid for cloning your vaccine construct.

9. Please revise the English language of your manuscript.

10. For the richness of the article, the authors should compare their study with other vaccine design studies. For this, the authors please use the following studies and cite them.

https://doi.org/10.2174/1573409919666230612125440

https://doi.org/10.1080/07391102.2023.2258403

https://doi.org/10.1007/s12033-023-00949-y

https://doi.org/10.1016/j.ijbiomac.2024.131517

6. PLOS authors have the option to publish the peer review history of their article (what does this mean? ). If published, this will include your full peer review and any attached files.

**Do you want your identity to be public for this peer review?** For information about this choice, including consent withdrawal, please see our Privacy Policy .

Reviewer #1: No

Reviewer #2: No

---

## [Author Response · Author response to Decision Letter 0]

10 Dec 2024

Response Letter to Editor (PLOS ONE)

Submission ID. PONE-D-24-40708

Title: Targeting HIV-1 Conserved Regions: An Immunoinformatic Pathway to Vaccine Innovation for the Asia

Journal: PLOS ONE

Respected Editor,

The authors gratefully acknowledge the editor and reviewers for spending valuable time reviewing the manuscript and making constructive comments and suggestions. We endeavored to fulfill all comments by the suggestions of the peer reviewers to the best of our abilities. We also believe that having followed your comments, the scientific and technical quality of the paper has been improved and fulfills the publication requirements of your esteemed journal, “PLOS ONE.”

We have added an updated figure for better visualization.

We are looking forward to hearing from you.

Faithfully,

Akmal Zubair, PhD

(On behalf of the Authors)

Reviewer #1: The authors have conducted the study ‘Targeting HIV-1 Conserved Regions: An Immunoinformatic Pathway to Vaccine Innovation for the Asia ” via comprehensive bioinformatics approaches, which makes for a valuable vaccine candidate. The results would introduce a vaccine candidate that could be taken to in vivo and possibly clinical studies. However, before considering it for publication, several points should be considered.

Major comments:

1. The abstract should be a little more detailed. It does not clearly describe steps that assess the immunogenic capability of the vaccine construct. In addition, there is no conclusion at the end of the Abstract.

Answer. The abstract has been updated and structured PAGE (1 and 2) the last line in the abstract is the conclusion part.

2. In the last paragraph of the Introduction, there was no hypothesis for this study, the authors should clarify the aims and hypothesis of this study.

Answer. The aim of this study is to design a novel stable in-silico vaccine for HIV that can the majority region of Asia. There are several in-silico vaccines developed for HIV, but all these vaccines use a single protein or partial protein as a vaccine candidate. We came up with the novel idea that weather-conserved regions in HIV genomics can act as stable vaccine candidates. For this purpose, the conserved region in the HIV genome was identified and used as an HIV vaccine candidate. (Page 4 lines 104-108)

3. What’s the difference between section 2.11 (Simulation-based computational approach) and section 2.14 (immune simulations)?

Answer Both are the same. I forgot to remove section 2.11. We have removed section 2.11 and the new section is under the heading (Simulation-based computational approach2.14)( page 9 line 247)

4. There are several published papers (PMID: 29571972, PMID: 31606417, PMID: 39331650, PMID: 37226084) on multi-epitope designed against HIV, providing advantages of your proposed vaccine.

Answer. There are several Insilco vaccines designed for HIV. These vaccines all utilize a single protein-based approach for vaccine development. Due to the high rate of mutations in the HIV genome, these vaccines may be compromised upon the mutation of that region, but in our vaccine structure, we used the conserved region in the HIV genome, which is less prone to mutation and possibly sustained for a longer period. (Page 4 lines 99-104) (also table 1 page 11)

Gene Name Conserved regions and sequences Position in Genome Size

Gag region GGAGCCACCCCACAAGATTTAAA

ATAGCAGGAACTACTAG

AAAATAGTAAGAATGTATAGCCCT

AGACAGGCTAATTTTTTAGG

TCCCTCAAATCACTCTTTGGCA 1568 to 1590

1742 to 1758

1850 to 1873

2324 to 2343

2528 to 2549 22 bp

16 bp

23 bp

19 bp

21 bp

Pol ACAGGAGCAGATGATACAGT

CCAATTAGTCCTATTGA

CCAGTAAAATTAAAGCCAGG

TTAAACAATGGCCATTGACAGAAGA

AAATCAGTAACAGTACT

CCACAGGGATGGAAAGG

AGCATGACAAAAATCTT

AGCTGGACTGTCAATGA

TGGACATATCAAATTTATCA

GTCAATACCCCTCCT

TATGCATTAGGAATCATTCA

TCATGGGTACCAGCACA

TGTGATAAATGTCAG 2605 to 2624

2827 to 2843

2851 to 2870

2888 to 2912

3136 to 3152

3274 to 3290

3313 to 3329

3577 to 3593

3835 to 3854

4075 to 4089

4329 to 4348

4428 to 4444

4626 to 4640 19 bp

16 bp

19bp

24 bp

16 bp

16 bp

16 bp

16 bp

19 bp

14 bp

19 bp

16 bp

14 bp

Vif ATCCCAGCAGAAACAGG

GGAATTCCCTACAATC

ATGGCAGTATTCATTCACAATTTTAA

AAGAAAAGGGGGGATTGGGGGGTACAGTGCAGG

GAAAGAATAATAGACAT 4773 to 4789

4926 to 4942

5040 to 5098

5100 to 5116 16 bp

16 bp

58 bp

16 bp

VPR CAGGGACAGCAGAGA

ATTTGGAAAGGACCAGC

TGGAAAGGTGAAGGGGCAGT 5189 to 5203

5208 to 5224

5235 to 5254 14 bp

16 bp

19 bp

Tat AGCAGAATAGGCATT

TCCTATGGCAGGAAGAAGCGGA 6072 to 6086

6253 to 6274 14 bp

21 bp

Env AATTGGAGAAGTGAA

GGAAGCACTATGGGCGC

GTCTGGGGCATTAAACA 8115 to 8129

8265 to 8281

8394 to 8410 14 bp

16 bp

16 bp

Nef GGATCAACAGCTCCT

CTCATCTGCACCACTA

CTTTTTAAAAGAAAAGGGGGGACTGGA 8450 to 8464

8490 to 8505

9591 to 9617 14 bp

15 bp

26 bp

5. It is not clear why the authors selected TLR3 and TLR5 for the docking of the vaccine construct. Please provide the rationale for using this receptor molecule.

Answer. TLR receptors are a crucial part of innate immunity to recognize pathogens. These are used as primary targets in-silico vaccine design to enhance the vaccine efficacy by triggering an immune response. Our HIV vaccine candidate can bind to TLR3, TLR5, TLR7, TLR8, and TLR9 but they show strong affinity to the TLR3 and TLR5 (page 24 lines 445-447 and 450-451)

6. The authors should provide the affinity between vaccine and TLR3 -5, which could show how the vaccine binds to TLR3 or TLR5. Perform the MD simulation with GROMACS trajectory and explore carefully to ensure the vaccine consistently binds to TLR3 or TLR5.

Answer. During docking with TLR-3 and TLR-TLR-5, the (Gag-Pol-Vpr-tat-Nef), vaccine construct achieved a minimum energy level of -336.17 and -376.66, respectively. The confidence scores for TLR3 and TLR8, respectively, are 0.9894 and 0.9764 as represented in Figure 5A and 5B. (pages 39- 44 line no 695-755)

Vaccine-TLR3 Complex MD Simulations

To evaluate the stability of the vaccine-TLR3 complex, a 100 ns molecular dynamics (MD) simulation was conducted using GROMACS, provide valuable insights into the stability and dynamics of this significant biomolecular interaction. The analysis involved multiple techniques, including root mean square deviation (RMSD), root mean square fluctuation (RMSF), radius of gyration (Rg), and solvent-accessible surface area (SASA), each contributing to a comprehensive understanding of the complex's behavior over time. The RMSD analysis revealed critical information regarding the system's stability. The observed plateau in the RMSD plot (Fig. A) indicated that the system reached equilibrium, confirming that the simulation duration was adequate for capturing the dynamics of the vaccine-TLR3 interaction. This stability is further supported by the absence of significant fluctuations, suggesting that the complex remains stable under dynamic conditions. Such findings are essential, as they imply that the vaccine maintains its structural integrity throughout the simulation, which is vital for effective immune engagement. In terms of flexibility, the RMSF analysis provided residue-specific fluctuation data, highlighting areas of higher mobility typically associated with flexible regions such as loops and coils in the ligand structure (Fig. B). These fluctuations are indicative of regions that may adapt during binding interactions, allowing for optimal fit and enhanced binding affinity. Understanding these flexible regions can aid in designing more effective vaccines by targeting specific residues that contribute to receptor recognition. The SASA analysis tracked variations in surface exposure over time, showing a decrease in SASA (Fig. C). This decrease suggests enhanced receptor-ligand interactions and indicates that as the simulation progressed, the complex adopted a more compact structure. This inverse relationship with RMSD indicates steady binding and structural stability between the multi-epitope vaccine and TLR3, reinforcing the idea that strong interactions lead to a more stable conformation conducive to immune activation. Additionally, the Rg plot (Fig. D) highlighted the compactness of the protein complex, with an average radius of gyration of approximately 3.6 nm. This measurement reflects a tighter atom distribution around the center of mass, further supporting the notion that robust inter-component interactions are established throughout the simulation period. A compact structure is critical for maintaining functional integrity and enhancing receptor-ligand interactions. These results collectively demonstrate that the vaccine-TLR3 complex is structurally stable and compact throughout the 100 ns MD simulation. The robust inter-component interactions observed suggest a favorable conformation for effective immune response activation. These findings highlight not only the potential efficacy of multi-epitope vaccines but also underscore the importance of molecular dynamics simulations in understanding biomolecular interactions. Future experimental validations will be essential to confirm these computational predictions and assess their implications for vaccine development against various pathogens.

Figure: The ligand-receptor complex (vaccine and TLRs) as simulated by molecular dynamics. (A) The docked complexes' stability throughout time is shown by the RMSD (Root Mean Square Deviation) study. (B) The peaks in the RMSF (Root Mean Square Fluctuation) plot show areas of high flexibility. (C) Information about the vaccine construct's surface exposure during the simulation is provided by the SASA (Solvent Accessible Surface Area) analysis. (D) The vaccine design sustains a stable, compact shape during the simulation period, as shown by the Rg (Radius of Gyration) plot.

MMPBSA binding free energy analysis

The thermodynamic variable known as the free energy of binding (ΔGbind) is thought to be crucial for evaluating the favorable protein-protein interaction and its affinity for precise biological system modeling. In this context, the free binding energy of the MD simulations was determined using the g_mmpbsa tool. Solvent-accessible surface area (SASA) and unfavorable polar solvation energy (PSE), two of the previously mentioned favorable forces, are computed by MM/PBSA. The MM/PBSA calculated free energy of binding for the system was approximated as −13950 ±15300 kJ/mol. The docking was energetically possible, as demonstrated by the negative values of Gibbs free energy (ΔG) in the data. A potential vaccine candidate, the observed negative free binding energy value shows that the vaccination complex is firmly attaching to the receptor.

Figure: MM/PBSA Analysis of Predicted Vaccine-Protein Interactions: Evaluating Binding Free Energies to Assess Stability and Efficacy of Designed Vaccine Candidates.

Reviewer 2 comments

Reviewer #2: 1. Please perform the molecular docking between T-cell epitopes and MHC molecules.

Answer. Methodology section (page 8 line 235-240))

Prediction of the 3D structure of T-cell epitopes and Molecular docking with HLA.

To perform molecular docking between HLA alleles and T-cell epitopes, the https://datascience.unm.edu/tomcat/biocomp/convert tool was used to convert the T-cell epitopes into pdb format. The respective HLA allele was retrieved from the protein data bank. Discovery Studio performed a refinement step before docking. The HDOCK server was used for molecular docking between T-cell epitopes and HLA alleles (page 8 line 235-240)

Result section. (page 24-27 lines 461-501)

The structure-based molecular docking was performed between the Helper T-cell epitopes and MHC-ii HLAs to determine the docking and confidence score by online HDOCK server. Figure 10 (a, b) illustrates the docking between Helper T-cells (HTLs) epitopes, (ERIIDIRDSREFGKD, CRERIIDIRDSREFG) and MHC-II HLA-DRB1*04:05 with binding-757.24, -229.92 with confidence score 1.00001 and 0.8318. similar in Figure 10 (c, d, e): docking between HTLs epitopes (GISYGRKKRKLEKSR, ISYGRKKRKLEKSRK, RIGISYGRKKRKLEK) with HLA-DRB1*13:02 having docking score -756.24, -840.70, -801.34 with confidence score 1.0000, each HTLs epitopes. (f, g). depict the interaction between the HLA-DRB1*15:01 receptor and the Helper T-cell epitope (TRNKVRMYSPRQANF, RNKVRMYSPRQANFL). The confidence scores were -717.40 and -721.79, and the confidence score was 1.0000 for T-cell epitopes with HLAs.

An illustration of the molecular docking of the cytotoxic T-cell epitopes (CTLs), FLGPSNHSL with the MHC-I allele HLA-A*02:01 may be seen in Figure 2(a) with the docking score -523.75 and confidence score -0.9994. Figure 2b illustrates the docking that takes place between QSIPLLMHY and HLA-B*15:01 with docking and confidence scores of -595.01 and .0999. Figure 2 (c,d) illustrates the molecular docking that occurs between LYTRNYYKI and MYSPRQANF when HLA-A*24:02 with confidence scores -605.10, -506.29, and 0.999, 0.9992. Similar to the , the docking between ILYTRNYYK and HLA-A*03:01 (-573.98, 0.9998), KSYLDCQSW and HLA-B*57:01 (-543.50, 0.996), and ISNSTPQGW and HLA-B*58:01 (-501.03, 0.9991) is shown in figure 2(e, f, g).

The figure represents the molecular docking between T-cell epitopes and MHC molecules. The T-cell epitope is represented by red colour while the MHC alleles are represented in cyan colour. (a, b). represent the docking between T-cell epitopes (ERIIDIRDSREFGKD, CRERIIDIRDSREFG) with HLA-DRB1*04:05 (c, d, e) docking between T-cell epitopes (GISYGRKKRKLEKSR, ISYGRKKRKLEKSRK, and RIGISYGRKKRKLEK with HLA-DRB1*13:02. (f, g). represent the docking of T-cell epitope (TRNKVRMYSPRQANF, RNKVRMYSPRQANFL) with HLA-DRB1*15:01.

Figure 2 represents the molecular of T-cell epitopes with MHC-1 class HLA alleles. Figure 2(a) represents the molecular docking of T cell epitope FLGPSNHSL with allele HLA-A*02:01. Figure 2b represents the docking between QSIPLLMHY and HLA-B*15:01. The figure 2 (c,d) shows the molecular docking between LYTRNYYKI and MYSPRQANF with HLA-A*24:02. Similarly, the figure 2(e, f g) represents the docking between ILYTRNYYK and HLA-A*03:01, KSYLDCQSW and HLA-B*57:01 and ISNSTPQGW with HLA-B*58:01.

2. Authors should evaluate the vaccine construct's antigenicity, allergenicity, and solubility.

Answer. The antigenicity of our vaccine construct is 0.782615, and the allergenicity score is

-1.290302. The solubility results revealed that it is well soluble in water. The solubility score is 0.995973. (Page 15 lines 337-339)

3. In the manuscript, the authors should provide a logical reason for using this type of linker in the structure of the vaccine by citing a valid reference.

Answer. We used different linkers such as GPGPG, KK, and AAY. Linkers such as GPGPG and KK were used to enhance the immunogenicity of vaccines. The linker AAY was used to reduce steric hindrance and allow the epitopes to properly fold. Validate references were added. (Page 17 lines 363-364)

4. Authors should predict discontinuous B-cell epitopes on the 3D structure of the vaccine.

ANSWER. we have performed the discontinuous epitopes prediction of B cells on 3D structure of Vaccines.( Page 13 lines 305 and figure 2)

5. The authors should present the results of the "Molecular Dynamic Simulation" and "Immune Simulation" sections in more detail.

Answer. MD simulation was further explained in detail. ((pages 39- 44 line no 695-755)

6. The authors should further discuss why the current vaccine constructs is more advantageous than the previous ones. How do you see the future steps for your vaccine? What are your next planned steps? Add a discussion regarding the next needed steps, Limitation, and etc.?

Answer. There are several Insilco vaccines designed for HIV. These vaccines all utilize a single protein-based approach for vaccine development. Due to the high rate of mutations in the HIV genome, these vaccines may be compromised upon the mutation of that region, but in our vaccine structure, we used the conserved region in the HIV genome, which is less prone to mutation and possibly sustained for a longer period. (page 46- 48, lines 837-882)

in silico vaccine design to be useful for human use, it must go through several phases of the systematic, regulated preclinical and clinical trial process before e

---

## [Decision Letter · Decision Letter 1]

27 Dec 2024

Targeting HIV-1 Conserved Regions: An Immunoinformatic Pathway to Vaccine Innovation for the Asia

PONE-D-24-40708R1

Dear Dr. Zubair,

We’re pleased to inform you that your manuscript has been judged scientifically suitable for publication and will be formally accepted for publication once it meets all outstanding technical requirements.

Kind regards,

Khalid Muhammad

Academic Editor

PLOS ONE

Additional Editor Comments (optional):

Reviewers' comments:

Reviewer's Responses to Questions

**Comments to the Author**

1. If the authors have adequately addressed your comments raised in a previous round of review and you feel that this manuscript is now acceptable for publication, you may indicate that here to bypass the “Comments to the Author” section, enter your conflict of interest statement in the “Confidential to Editor” section, and submit your "Accept" recommendation.

Reviewer #1: All comments have been addressed

Reviewer #2: All comments have been addressed

2. Is the manuscript technically sound, and do the data support the conclusions?

Reviewer #1: Yes

Reviewer #2: Yes

3. Has the statistical analysis been performed appropriately and rigorously? 

Reviewer #1: N/A

Reviewer #2: N/A

4. Have the authors made all data underlying the findings in their manuscript fully available?

Reviewer #1: Yes

Reviewer #2: Yes

5. Is the manuscript presented in an intelligible fashion and written in standard English?

Reviewer #1: (No Response)

Reviewer #2: Yes

6. Review Comments to the Author

Reviewer #1: The authors have responded appropriately to my comments, and the article is acceptable in this form.

Reviewer #2: The authors took all comments into consideration and I hope this paper will be well received by readers.

7. PLOS authors have the option to publish the peer review history of their article (what does this mean? ). If published, this will include your full peer review and any attached files.

**Do you want your identity to be public for this peer review?** For information about this choice, including consent withdrawal, please see our Privacy Policy .

Reviewer #1: No

Reviewer #2: No

---

## [Editor Report · Acceptance letter]

PONE-D-24-40708R1

PLOS ONE

Dear Dr. Zubair,

I'm pleased to inform you that your manuscript has been deemed suitable for publication in PLOS ONE. Congratulations! Your manuscript is now being handed over to our production team.

Kind regards,

on behalf of

Dr. Khalid Muhammad

Academic Editor

PLOS ONE